# CD28-deficient mice are vulnerable to mouse papillomavirus MmuPV1 infection of the skin and mucosae

Sarah Brendle[1,2�], Jingwei Li[1,2☍], Song Lu[2☍], Todd D. Schell[3☍], Michael Kozak[1,2☍], Vonn Walter[4,5], Debra Shearer[1,2], Joshua Place[6], Karla Balogh[1,2], Jean-Laurent Casanova[7,8,9,10,11], Neil Christensen[1,2,3], Adam D. Burgener[12,13,14], Thomas T. Murooka[15], Yusheng Zhu[2], Vivien Béziat[7,8,9,10], Jiafen Hu [1,2]*

1 Jake Gittlen Laboratories for Cancer Research, Pennsylvania State University, Hershey, Pennsylvania, United States of America, 2 Department of Pathology and Laboratory Medicine, Pennsylvania State University, Hershey, Pennsylvania, United States of America, 3 Department of Cell & Biological Systems, Pennsylvania State University, Hershey, Pennsylvania, United States of America, 4 Department of Public Health Sciences, Pennsylvania State University, Hershey, Pennsylvania, United States of America, 5 Department of Molecular and Precision Medicine, Pennsylvania State University, Hershey, Pennsylvania, United States of America, 6 Department of Comparative Medicine, Pennsylvania State University, College of Medicine, Hershey, Pennsylvania, United States of America, 7 Laboratory of Human Genetics of Infectious Diseases, Necker Branch, INSERM, Necker Hospital for Sick Children, Paris, France, 8 Imagine Institute, Paris-Cité University, Paris, France, 9 St. Giles Laboratory of Human Genetics of Infectious Diseases, Rockefeller Branch, The Rockefeller University, New York, New York, United States of America, 10 Pediatric Hematology-Immunology Unit, Necker Hospital for Sick Children, AP-HP, Paris, France, 11 Howard Hughes Medical Institute, New York, New York, United States of America, 12 Center for Global Health and Diseases, Department of Pathology, School of Medicine, Case Western Reserve University, Cleveland, Ohio, United States of America, 13 Department of Obstetrics, Gynecology, and Reproductive Sciences, University of Manitoba, Winnipeg, Canada, 14 Department of Medicine, Solna, Unit of Infectious Diseases, Center for Molecular Medicine, Karolinska Institutet, Stockholm, Sweden, 15 Department of Immunology, University of Manitoba, Manitoba, Winnipeg, Canada

☍ These authors contributed equally to this work as first authors.
* fjh4@psu.edu

## Abstract

CD28 is a co-stimulatory molecule expressed on the surface of T cells. To date, three individuals with germline CD28 deficiency have been reported to develop recalcitrant, HPV-driven warts: one exhibited persistent lesions, another experienced disease resolution, and the third developed a chronic "tree-man" phenotype. In mice, we confirmed that CD28-knockout (CD28ko) animals on the C57BL/6 (B6) background are susceptible to cutaneous infection with mouse papillomavirus (MmuPV1); however, their skin warts regressed spontaneously approximately five weeks post-infection. Furthermore, we demonstrate that CD28ko mice are vulnerable to MmuPV1 infection at HPV-relevant mucosal sites, including the most HPV prevalent sites: anogenital tract and oral cavity. Virions recovered from vaginal lavage were infectious but could be neutralized by the neutralizing monoclonal antibody MPV.A4. Viral clearance at mucosal sites was delayed in CD28ko mice, persisting for up to six weeks in the lower genital tract. Blocking the CD28 ligands CD80 and CD86 in B6 mice reproduced the CD28ko phenotype following MmuPV1 infection and markedly

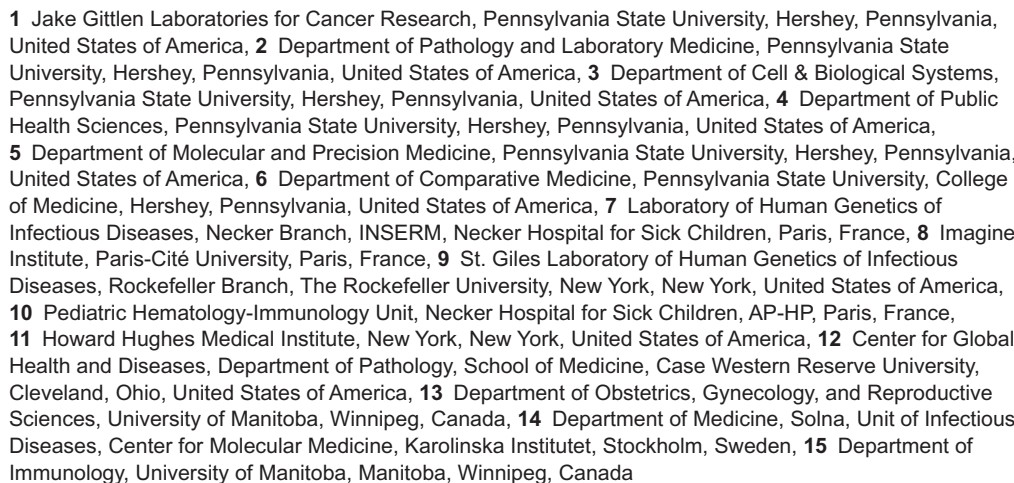

**Data availability statement:** All relevant data are in the manuscript and its Supporting information files.

**Funding:** o This work was supported by The National Institute of Health (NIH) (the Dental and Craniofacial Research program under Award Number 1R21DE028650 to JH; National Cancer Institute under Award Number 1 R21 CA271069 to JH and TTM), Canada Institute of Health Research (52250/322744 to TTM, ADB, and JH), the Jake Gittlen Memorial Golf Tournament (to JH and NDC), Penn State Cancer Institute Program Project Development Award Sponsored by Highmark Community Health Reinvestment Fund (to JH and NDC), and Pathology Department Research Funds (to JH). The funders had no role in study design, data collection and analysis, decision to publish, or preparation of the manuscript.

**Competing interests:** The authors have declared that no competing interests exist.

reduced CD28 expression, implicating the CD28-CD80/CD86 axis in delayed viral clearance. Infected CD28ko mice showed a reduction in both CD4$^+$ and CD8$^+$ T cell population in the spleen compared to infected B6 mice, but an increase in CD11c$^+$/F4-80$^+$ cells, particularly the plasmacytoid dendritic cell (pDCs, SiglecH$^+$) subset. Additionally, CD28ko mice exhibited delayed recruitment of activated CD4$^+$ T cells to infected tissues. Accumulation of MmuPV1 E6/90–99-specific, tetramer-positive CD8$^+$ cytotoxic T lymphocytes (CTLs) was slower in CD28ko than in B6 mice; these CTLs remained FoxP3 negative but displayed reduced efficacy in both in vitro killing and antiviral cytokine assays. Adoptive transfer of CTLs from either B6 or CD28ko mice into MmuPV1-infected Rag1ko mice induced viral clearance at mucosal (oral) sites, whereas B6-derived CTLs achieved more complete regression of cutaneous (tail) lesions. Collectively, these findings indicate that CD28 deficiency delays but does not prevent the clearance of papillomavirus infections at both cutaneous and mucosal sites in mice.

## Author summary

Human papillomaviruses (HPVs) are ubiquitous, and high-risk HPV types account for approximately 5% of all human cancers. Although prophylactic vaccines are available, no effective therapy exists for established HPV infections or their sequelae. Viral persistence is central to HPV pathogenesis, and individuals with primary or secondary immunodeficiencies exhibit elevated rates of HPV infection and dysplasia. Our previous work demonstrated that CD28-deficient patients develop recalcitrant HPV-driven lesions, including the extreme "tree-man" phenotype. In mice, CD28-knockout (CD28ko) animals develop MmuPV1-induced cutaneous warts with delayed viral clearance. Here, we investigated whether MmuPV1 can infect three HPV-relevant mucosal sites, the anal tract, genital tract and oropharynx, of CD28ko mice. All sites proved susceptible. To dissect CD28-dependent immune control, we blocked CD80/CD86-expressing cells in B6 mice; this recapitulated the CD28ko phenotype and delayed viral clearance at both skin and mucosa. Further depletion of CD4$^+$ and CD8$^+$ T cells in CD28ko mice prolonged viral persistence. CD28ko mice exhibited delayed expansion of CD8$^+$ T cells and impaired recruitment of CD4$^+$ T cells to infected tissues. MmuPV1 E6/90–99-specific CD8$^+$ cytotoxic T lymphocytes (CTLs) from infected B6 mice displayed greater in vitro cytotoxicity and antiviral activity than those from CD28ko mice. Adoptive transfer of these CTLs from either genotype into MmuPV1-infected Rag1ko mice induced tumor regression, but B6-derived CTLs were more effective, leading to complete clearance particularly at cutaneous sites. Collectively, these findings indicate that CD28 deficiency delays but does not prevent clearance of papillomavirus at both cutaneous and mucosal sites, likely through impaired T-cell recruitment and function. This study provides new insights into the immunological mechanisms underlying HPV pathogenesis

## Introduction

Human papillomaviruses (HPVs) are among the most common pathogens worldwide, infecting cutaneous and mucosal epithelia across diverse anatomical sites. High-risk HPV types are etiologically linked to approximately 5% of human cancers, including cervical, anogenital, and oropharyngeal carcinomas, as well as a subset of non-melanoma skin cancers [1]. Despite the availability of highly effective prophylactic vaccines, these interventions do not clear established infections or treat HPV-associated lesions, leaving millions of individuals at risk for persistent infection and malignant progression. Consequently, understanding the immunological mechanisms that govern viral clearance remains a critical priority for developing therapeutic strategies.

Viral persistence is central to HPV pathogenesis. Host immune responses are crucial for the control of HPV-associated infections [2–4]. In immunocompetent hosts, most infections resolve spontaneously; however, individuals with primary or secondary immunodeficiencies exhibit markedly increased susceptibility to chronic HPV infection and associated dysplasia [3–7]. Clinical observations underscore the importance of T-cell–mediated immunity in controlling papillomavirus infections. For example, patients with germline defects in co-stimulatory pathways, such as CD28 deficiency, develop HPV4 (γ-papillomavirus)–related severe, recalcitrant warts and HPV2 (α-papillomavirus)–related 'tree man syndrome' (TMS) [8–10] but these patients are normally resistant to other infections [10]. CD28 is a critical co-stimulatory receptor expressed on T cells, required for optimal activation, expansion, and survival of effector lymphocytes [11]. CD28 expression is required after T cell priming for intact effector CD4+ T cell responses during infection [11,12]. Engagement of CD28 by its ligands CD80 and CD86 amplifies T-cell receptor signaling, promotes cytokine production, and enhances cytotoxic function [11,13]. While its role in systemic immunity is well established, the contribution of CD28-dependent pathways to local immune control of papillomavirus infection at cutaneous and mucosal sites remains poorly defined.

Our previous work demonstrated that CD28-knockout (CD28ko) mice on the C57BL/6 (B6) background are susceptible to cutaneous infection with mouse papillomavirus (MmuPV1), developing warts that regress only after a significant delay compared to wild-type controls [10,14]. However, whether CD28 deficiency similarly impacts viral clearance at mucosal sites relevant to human HPV disease, such as the anogenital tract and oropharynx, has not been investigated. Previous studies have shown increased susceptibility to mucosal infection in select mouse strains compared to cutaneous sites [14–16]. Given the low number of patients with CD28 mutations and the potential for missed opportunities to closely monitor mucosal infections, using a mouse model is critical to address both viral clearance and associated immune responses. More critically, the mechanisms underlying the initial failure of CD28ko mice to control viral infections, in contrast to B6 mice, and their eventual ability to clear the virus are poorly understood. This question has important clinical implications. In patients, cutaneous lesions are readily identifiable, whereas mucosal infections may go undetected, potentially contributing to disease burden. Understanding whether CD28-mediated co-stimulation influences viral clearance at mucosal sites could reveal mechanisms underlying tissue-specific immune control and inform strategies for immunomodulation. This will help address several critical questions, including: (1) Are mucosal sites more susceptible to infection, as observed in other mouse strains? (2) Are T cells and other immune cells more important for viral control at mucosal sites than at cutaneous sites?

In this study, we addressed these gaps by examining the susceptibility of CD28ko mice to MmuPV1 infection at three HPV-relevant mucosal sites and by dissecting the immunological pathways involved in viral control. We further evaluated the impact of CD28 ligand blocking, T-cell subset depletion, and adoptive transfer of virus-specific cytotoxic T lymphocytes (CTLs) on infection outcomes. Our findings reveal that CD28 deficiency delays, but does not prevent, clearance of

papillomavirus at both cutaneous and mucosal sites, likely due to impaired functional T cells. These results provide new insights into the immunological mechanisms underlying HPV pathogenesis and highlight the critical role of CD28-dependent co-stimulation in antiviral immunity.

## Results

### CD28ko mice are susceptible to both cutaneous and mucosal infections

We previously reported that both male and female CD28ko mice were susceptible to MmuPV1 infection at two cutaneous sites (muzzle and tail) [10], consistent with findings in other mouse strains [17,18]. In this study, we further examined whether mucosal sites (oral, anal, and vaginal in females) were also susceptible to MmuPV1 infection, as demonstrated previously in other strains [16,19–22]. We infected four male and four female CD28ko mice at two cutaneous sites (tail and muzzle) and three HPV-relevant mucosal sites (vagina in females, anus and tongue for both males and females) and evaluated the mice at week four post-infection, a time point that detected active mucosal viral infections in previous studies [19,23]. Consistent with our previous report [10], visible lesions were detected in both male and female CD28ko mice around week three post-infection (Fig 1A). These lesions were positive for viral E4 protein at week 4 (Fig 1B). All simultaneously infected tissues were analyzed for viral RNA transcripts using qRT-PCR, normalized to total tissue RNA. Fig 1C shows viral transcript levels normalized to total RNA from different tissues harvested four weeks post-MmuPV1 infection (vaginal tissues were excluded from this comparison). Although direct comparison between cutaneous and mucosal tissues is limited by differences in scarification and tissue characteristics, these comparisons are relevant given prior reports of location-dependent infection severity in nude mice [23,24]. We observed slightly higher viral transcript levels in cutaneous sites than in mucosal sites, with anal tissue showing the lowest levels. Both cutaneous sites exhibited comparable viral RNA levels, and anal tissue contained significantly fewer transcripts than tail tissue at the same post-infection time point (Fig 1C, p<0.05).

Viral activity in infected tongue and anogenital tissues was further examined using in situ analyses, with representative images shown in Fig 1D–1I. The MmuPV1 E4 protein, previously identified as the most abundant viral protein produced post-infection [25,26], was detected in the infected anus (Fig 1D), tongue (Fig 1E), and vagina (Fig 1F, at week 2 and Fig 1G at week 4 post-viral infection) by immunohistochemistry (IHC) in infected CD28ko mice similarly to those observed in athymic mice [23,24] but not in B6 mice [16]. Viral DNA (Fig 1H) and viral RNA (Fig 1I) were also detected in the infected vaginal tissues of CD28ko mice. Tongue infections were localized at the base of the tongue (Fig 1E), consistent with our previous reports in other mouse strains [15,23,27,28], further confirming that MmuPV1 infection patterns are conserved across strains [21].

To further assess viral activity in the tongues of CD28ko mice, we infected 18 CD28ko mice (nine females and nine males) and 10 B6 mice (five females and five males) at the base of the tongue, as previously described [29]. All mice were sacrificed at week four post-infection, and infected tongues were analyzed for viral E1^E4 RNA [30]. Viral RNA levels were significantly higher in CD28ko mice compared to B6 mice (Fig 1J, p<0.01).

We monitored six B6, seven CD28ko, and three Rag1ko female mice for up to six weeks following lower genital tract infection [31]. At week 2 post-viral infection, all Rag1ko and CD28ko mice and five out of six B6 mice were positive for viral DNA (Fig 1K). Viral DNA levels were significantly higher in CD28ko mice compared to B6 mice through week five post-infection (Fig 1K, p<0.05). After week four post-infection, viral DNA levels were significantly higher in Rag1ko mice compared to both CD28ko and B6 mice (Fig 1K, p<0.05). The absence of detectable viral DNA in B6 mice after week four was reported in previous studies [15,16]. These findings indicate that viral infections in CD28ko mice persist longer at both cutaneous and mucosal sites compared to infections in B6 mice.

### Virus in cervicovaginal lavages of CD28ko mice is infectious

Lower genital tissues from CD28ko mice were harvested for viral protein detection, with Rag1ko female mice infected with MmuPV1 serving as positive controls. Positive viral L1 signals were detected in vaginal tissues of Rag1ko (Fig 2A)

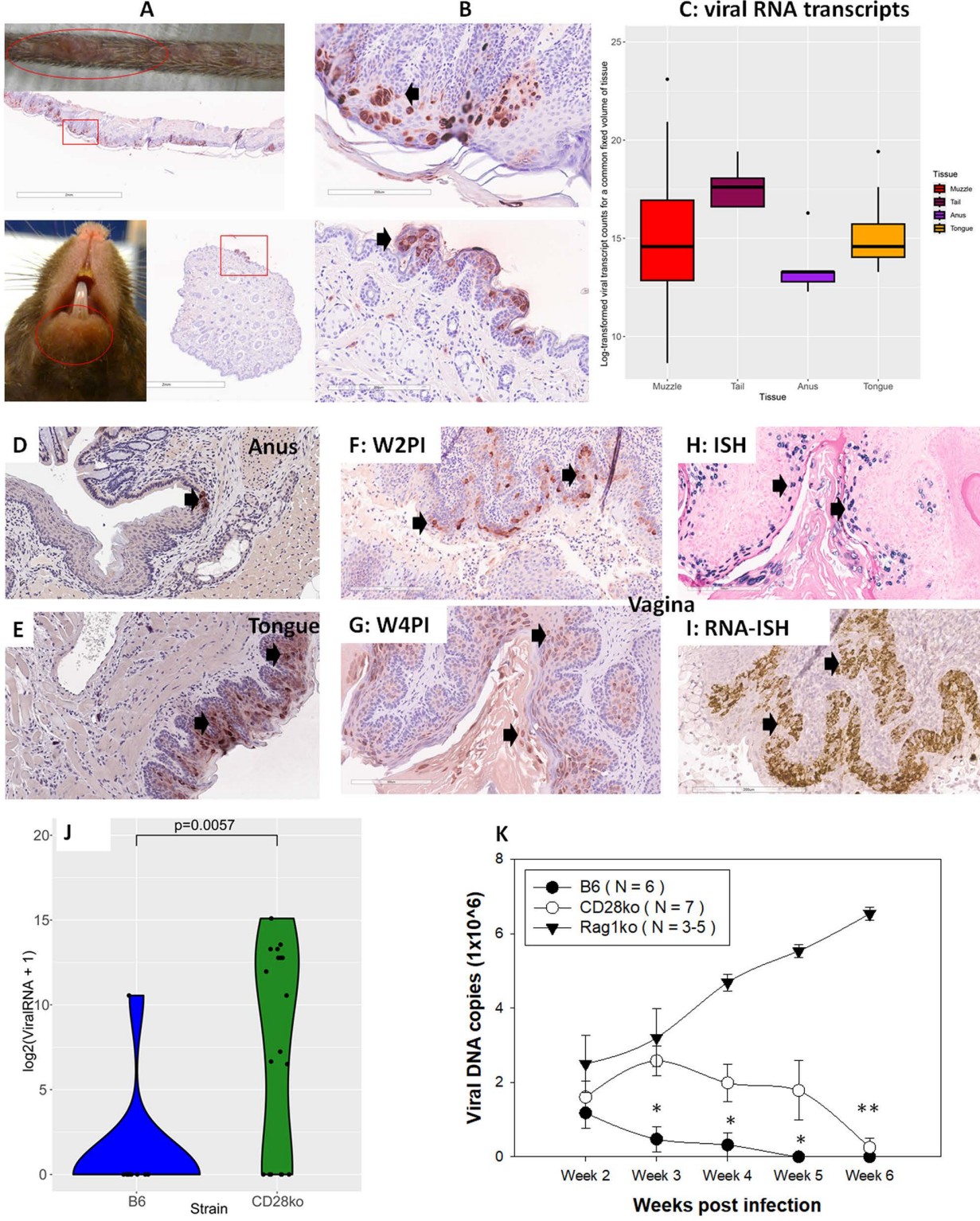

**Fig 1. CD28ko mice were susceptible to MmuPV1 infection at cutaneous and mucosal sites.** Both male and female CD28ko mice (N = 5/group) were infected with MmuPV1 simultaneously at two cutaneous sites (muzzle and tail) and three mucosal sites (vagina for females, anus, and tongue) and sacrificed at week 4 post infection. All infected tissues were harvested for viral transcript quantification and in situ assays for viral DNA, RNA, and protein

as reported previously. Visible lesions were detected at both tail and muzzle (A) around week three post viral infections. These lesions were found to be positive for MmuPV1 E4 protein by immunohistochemistry (B, arrows, IHC, signals in red). More viral RNA transcripts were detected in infected cutaneous tissues than in the infected mucosal tissues. Significantly more viral transcripts were found in the tail when compared to the anus (C, p<0.05). CD28ko mice were susceptible to MmuPV1 infection at mucosal (tongue, anus, and vaginal) sites. Compared to B6 mice, significantly higher levels of viral RNA were detected in infected tongues (C, p<0.05). MmuPV1 E4 protein was detected in anus **(D)**, tongue **(E)**, and vagina at week 2(F) and 4 (G) post viral infection. Viral DNA (H) and RNA (I) were detected by immunohistochemistry (IHC, signals in red), in situ hybridization (signals in blue, ISH), and in situ RNA hybridization (signals in brown, RNA-ISH) respectively. Significantly more viral RNA copies were found in the tongue of CD28ko mice when compared to B6 mice (J, p<0.05). Viral DNA was detected in the infected vaginal tract of CD28ko mice by qPCR until week 6 post viral infection while no viral DNA was detected in B6 mice at week 3 post viral infection (K, *p<0.05).

and CD28ko (**Fig 2B**) mice, indicating that infectious virions were produced in genital tissues. Infectious virus has previously been detected in cervicovaginal lavages of athymic mice and other mouse strains [23,32]. Virus isolated from infected CD28ko tissues was neutralized by the monoclonal antibody MPV.A4 but not by the isotype control H18.J4 [23,26,29,30,32] (**Fig 2C**, p<0.05). Additionally, Ki67 expression was observed in CD28ko tissues, indicating epithelial proliferation associated with infection (**S1 Fig**). The comparable titer of serum samples from both B6 and CD28ko infected mice neutralized MmuPV1 infection in vitro (**Fig 2D**, p<0.05), indicating that CD28ko mice mounted humoral immune responses comparable to those of B6 mice. These findings suggest that viral particles present in cervicovaginal lavages of CD28ko mice are infectious and potentially transmissible, consistent with observations in other mouse strains reported previously [23,33].

## CD28 ligands CD80/CD86 inhibition as well CD4 and CD8 depletion promoted tumor growth in CD28ko mice

Immune responses, particularly T cell–mediated immunity, play a critical role in controlling papillomavirus infections [34,35]. Previous studies have shown both CD4 and CD8 cells are required to control MmuPV1 infection, as evidenced by the absence of visible lesions [36,37]. CD28 ligands, CD80 and CD86, are essential for T cell activation during viral infection [11]. To assess whether CD80 and CD86 contribute to delayed viral clearance, we blocked CD80/CD86 ligands in B6 and CD28ko mice by administering monoclonal antibodies that specifically target CD80 and CD86, thereby preventing their interaction with CD28 and inhibiting costimulatory signaling [15]. In B6 mice, CD80/CD86 blockade resulted in a significant reduction in CD28 RNA levels in the local infected tissues, whereas no change was detected in CD28ko

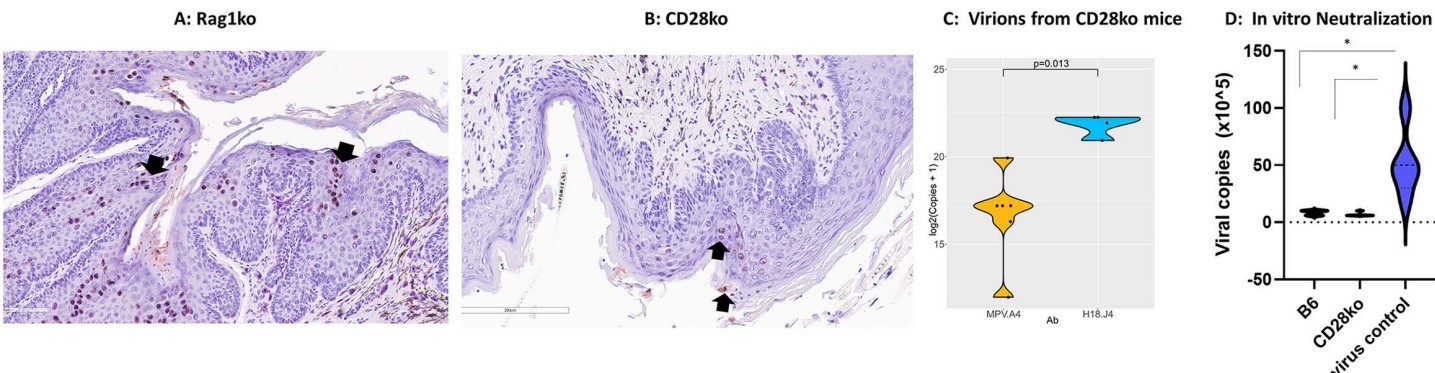

**Fig 2. Viral capsid L1 was detected in the infected vaginal tissues.** Tissues and cervicovaginal lavages from both Rag1ko mice (N = 5, A) and CD28ko mice (N = 5, B) were harvested for viral detection. Viral capsid L1 was detected by immunohistochemistry (IHC, signals in red). The cervicovaginal lavages were tested for infection and neutralization in K38 cells. The samples with infectious virions from MmuPV1 infected CD28ko mice could be neutralized by our monoclonal antibody MPV.A4 against L1 but not the isotype control H18.J4 (C, *p<0.05) suggesting infectious virions can be collected from CD28ko mice and these virions are assembled properly in the infected cells. **(D)** Sera from both B6 and CD28ko infected mice displayed a similar ability to neutralize viral copies through the in vitro neutralization assay, regardless of sex (*p<0.05).

mice (**Fig 3C**, p<0.05). The mean ImageJ intensity (based on five fields of view) of E4 protein was significantly higher in CD80/CD86-inhibited B6 mice compared to untreated B6 controls (**Fig 3D**, p<0.05) but was comparable to that observed in CD28ko mice (**Fig 3D**, p>0.05). Visible tail lesions were found in CD80/CD86 inhibited B6 mice and CD28ko mice at week three post viral infection (**Fig 3B**, top and bottom panel respectively). These findings suggest that blocking of CD80/CD86-expressing cells in B6 mice leads to a loss of CD28 expression in the infected tissues, increased viral activity resulting in tumor persistence, and mimics the delayed viral clearance seen in CD28ko mice. In CD28ko mice, no additional effect was detected following CD80/CD86 blocking due to the absence of CD28–ligand interaction. Therefore, the CD28–CD80/CD86 co-stimulatory pathway plays a critical role in controlling viral persistence.

Papillomavirus infections are primarily controlled by T cell–mediated immune responses, with both CD4+ and CD8+ T cells playing essential roles in viral clearance. Previous studies have demonstrated that depletion of CD4+ T cells

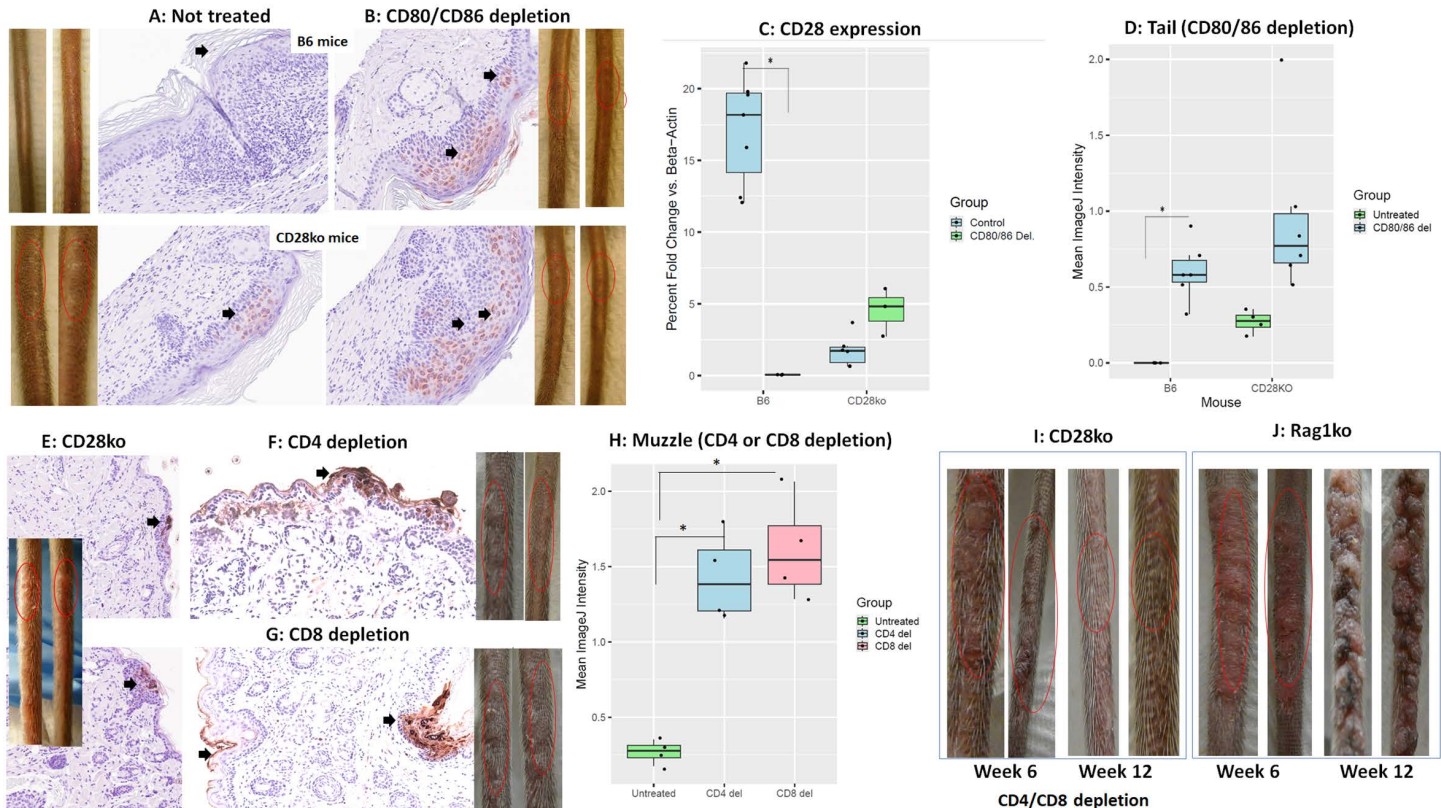

**Fig 3. Larger lesions and higher viral loads were observed in CD80/CD86-depleted B6 mice compared to untreated B6 mice.** Representative tail lesions and MmuPV1 E4 IHC were from control (N = 10, **A**), CD80/CD86 depleted B6 and CD28ko (N = 4/group, B, **D**), CD4 (N = 3, **E**), CD8 (N = 3, F) or CD4/CD8 (N = 5) depleted **(H)** CD28ko and Rag1ko (I) mice. Tail lesions were visible in infected tails of CD28ko mice (A, the bottom panel and D) with viral E4 positivity (20x, signals in red, arrows) but no lesions and few E4 positive cells (20x, signals in red, arrows) were found in B6 mice (A, the top panel) at week three post MmuPV1 infection. Visible tail lesions were found in CD80/CD86 depleted B6 mice and CD28ko mice (B, top and bottom panel respectively). Significantly increased mean ImageJ intensities (calculated from five field views) of MmuPV1 E4 protein in tail lesions were found in both B6 and CD28ko after CD80/CD86 blocking compared to corresponding untreated infected B6 and CD28ko mice (C, *p<0.05). Increased disease burden manifested as larger tumor sizes on the tails of CD28ko mice with CD4 (E) or CD8 (F) depletion, and we also detected increased mean imageJ intensities (calculated from five field views) of MmuPV1 E4 staining in the muzzle lesions of CD4 and CD8 depleted mice compared to untreated CD28ko mice (G, *p<0.05). Persistent tail lesions with increased size were detected in CD4/CD8 dual depleted CD28ko mice (H) although lesions were smaller than those in immunocompromised Rag1ko mice (I) especially at week 12 post infection. At week 6, the size of visible lesions in CD4/CD8 depleted CD28ko mice was comparable to Rag1ko mice. All lesions regressed after termination of CD4/CD8 depletion in these CD28ko mice.

results in greater susceptibility to MmuPV1 infection than depletion of CD8+ T cells, highlighting the critical role of CD4+ T-cell–dependent immunity [15,16,36,37]. CD28 signaling is a key co-stimulatory pathway required for optimal activation and function of CD4+ T cells, mediated through interactions with its ligands CD80 and CD86 on antigen-presenting cells. Although CD28-deficient mice exhibit increased susceptibility to MmuPV1 infection, they ultimately clear the virus, suggesting that CD28 signaling accelerates viral clearance but is not absolutely required [17,36,37]. CD28 signaling is essential for multiple facets of CD4+ T cell activation [12]. Next, we examined whether depletion of CD4+ (**Fig 3F** or CD8+ (**Fig 3G**) T cells promotes viral persistence and lesion growth in CD28ko mice. Tail lesions of CD4+ or CD8+ -depleted CD28ko mice were larger than those in the untreated CD28ko mice. Compared to untreated CD28ko mice (**Fi. 3E,** N = 3/group), increased intensity of E4 staining (based on five fields of view) was detected in muzzle lesions of both CD4 and CD8 depleted CD28ko mice compared to untreated CD28ko mice (**Fig 3H**, p < 0.05). We also tested CD4/CD8 double depletion in some CD28ko animals (**Fig 3I**). Persistent tail lesions developed in these mice and were visible at week 12 post-infection; however, tail lesion size was significantly smaller than that observed in Rag1ko mice (**Fig 3J**). Interestingly, these lesions regressed after cessation of CD4/CD8 double depletion, similar to the effect observed in a rabbit model when the T-cell suppressor cyclosporine was withdrawn following persistent infection established by a regressive construct [38]. These findings indicate that CD4 and CD8 T cells, individually and in combination, play critical roles in controlling viral clearance in CD28ko mice.

## Delayed CD4 positive immune cell infiltration in infected tissues of CD28ko mice

The correlation of papillomavirus-associated lesions with increased immune cell infiltration including T cells and other immune cells has been demonstrated in human studies and other preclinical PV models [39–47]. To further define the role of CD4+ T cells in CD28-dependent and independent pathways during delayed tumor clearance in the local tissues, we monitored infected tails over time in B6 and CD28ko mice. In addition to CD4 antibody staining, we co-stained these tissues using a T cell receptor (TCR)-specific antibody [48]. TCR beta (β) is part of the functional TCR heterodimer expressed at the cell surface of active αβCD3 T cells. CD4/TCRβ cells appeared near the infected epithelial layer as early as week 2 post-infection in B6 mice (**Fig 4A**). In contrast, a delay of CD4 T cell infiltration to the infected epithelial layer was observed in CD28ko mice with CD4 T cells not arriving until week 4 post infection (**Fig 4B**). Although we did not stain for MmuPV1 markers on these frozen sections, we performed RNA-ISH to detect viral RNA transcripts on FFPE tissues from these mice. The results show viral RNA was absent in infected tissues of B6 mice (**Fig 4C**, top panel) while abundant viral E4 transcripts were detected in infected tissues of CD28ko mice at week 4 post infection (**Fig 4C**, arrows, bottom panel). Previous studies have demonstrated that papillomavirus preferentially resides within the hair follicles, where stem cells are located [49]. While we cannot confirm that T cells localized to the hair follicles in our images, the presence of T cells in this niche may suggest a direct interaction contributing to the clearance of virus-infected cells in B6 mice. The role of these T cells in viral control warrants further investigation.

We next assessed whether there were differences in the numbers of other innate immune cells, particularly antigen-presenting cells such as macrophages and dendritic cells, in the infected tail tissues of B6 and CD28ko mice at three weeks post-infection using mean fluorescence intensity (MFI) based on five field views (**S3 Fig**). B7-1 (CD80+) and B7-2 (CD86+) are expressed on various APCs and serve as critical costimulatory molecules for T-cell activation through interaction with CD28. Notably, CD86 exhibits a higher affinity for CD28, while CD80 preferentially binds CTLA-4. These interactions are essential for shaping T-cell responses during infection and immune regulation. CD86 signals observed in the epithelium and underlying tissues were quantified because these cells represent the recruitment of antigen-presenting cells that may compensate for the absence of CD28 expression in CD28ko mice. This analysis provides insight into potential compensatory immune mechanisms involved in viral control. Regions of interest included epithelial and subepithelial compartments, and signal intensity was measured after applying thresholding and masking to exclude nonspecific background staining. Comparable levels of macrophages (F4-80+) were observed in the infected tissues of B6 and CD28ko

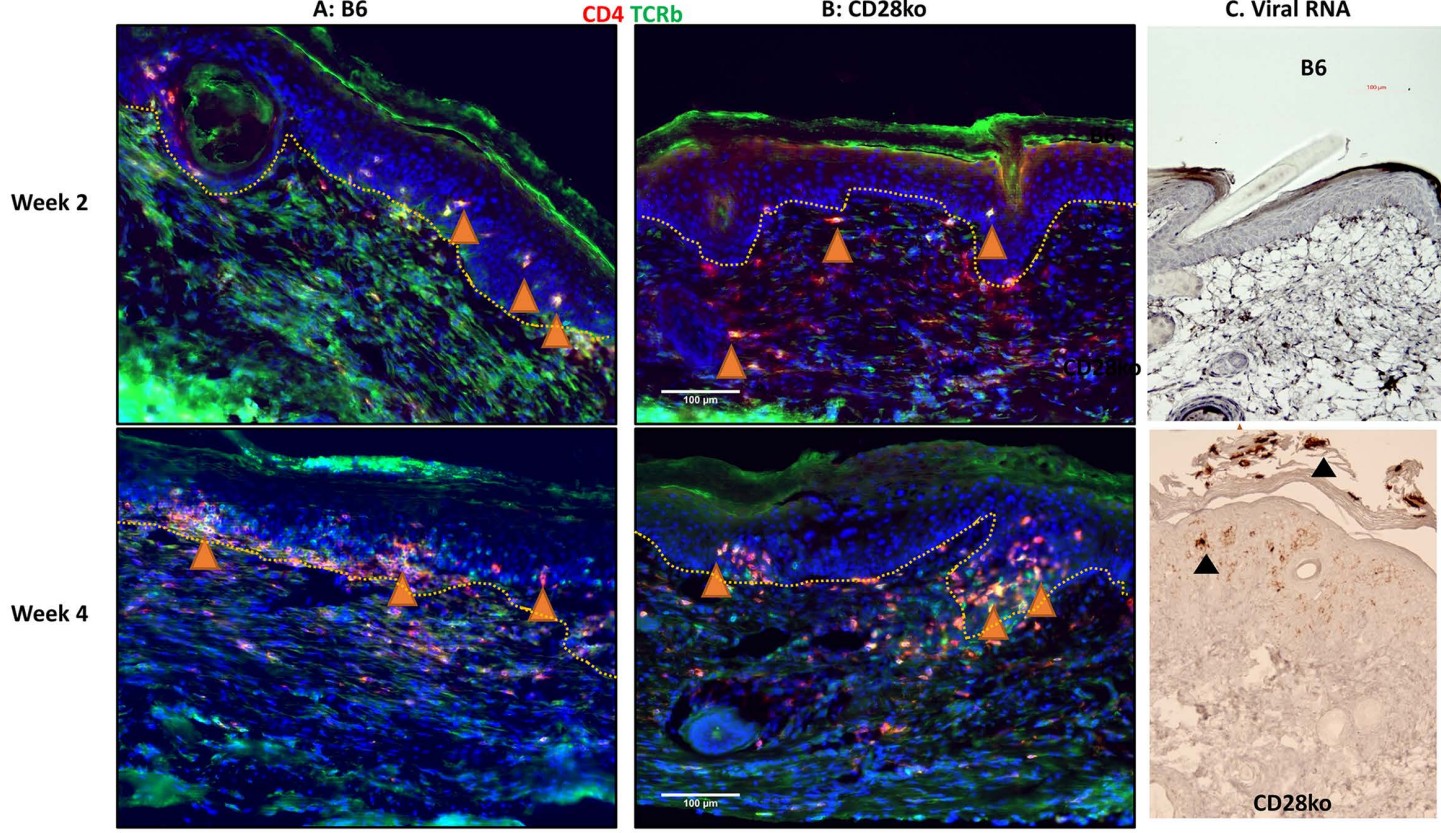

**Fig 4. Infiltration of active CD4 T cells to the infected epithelium was delayed in the infected tails of CD28ko mice.** TCRβ positive CD4 T cells appeared near the infected epithelial layer of infected tails as early as week 2 in B6 mice (N = 3, **A)**. In contrast, a delay of CD4 T cell infiltration to the infected epithelial layer was observed in CD28ko mice until week 4 post infection (N = 3, **B)**. Viral signals (arrows) were detected in CD28ko tissues by RNA-ISH but absent in B6 tissues (C) at week 4 post infection suggesting complete elimination of MmuPV1 infected cells in B6 mice. Infiltration of myeloid cells including macrophages (F4-80+), dendritic cells (CD11c), and B7-2 (CD86) in the infected tail of B6 and CD28ko mice at three weeks post infection is shown with Mean fluorescent intensity (MFI) calculated from five field views using ImageJ software. Comparable numbers of macrophages (F4-80+) were found between these two mouse strains (D, n.s. p > 0.05). However, significantly higher numbers of CD11c+ and CD86+ cells were identified in infected tail tissues of CD28ko mice when compared to B6 mice (E and F, *p < 0.05).

mice (**S3B Fig**, p > 0.05). However, significantly higher numbers of antigen-presenting cells expressing B7-2 (CD86+) (**S3C Fig**, p < 0.05) and dendritic cells (CD11c+) (**S3D Fig**, p < 0.05) were detected in the infected tail tissues of CD28ko mice compared to B6 mice.

### Changes in different immune cell populations in the spleens of CD28ko infected mice

We next used multicolor flow cytometry to assess differences in several representative immune cell populations [19,50] in the spleens of uninfected control (N = 11; both males and females) and infected CD28ko mice (N = 15; both males and females), and infected B6 (N = 22; both males and females) (**Fig 5**). Significantly lower levels of CD4+ and CD8+T cells were observed in infected CD28ko mice compared to uninfected CD28ko mice (**Fig 5A**, p < 0.05). Comparable CD4+T and CD8+T cells were observed in infected CD28ko mice compared to infected B6 mice (**Fig 5B**, p > 0.05). In contrast, we observed increased frequencies of CD11c+F4-80+double-positive cells in infected CD28ko mice compared to B6 mice (**Fig 5B**, p < 0.05). Within this population, the plasmacytoid dendritic cell subset (pDCs, SiglecH+) was significantly

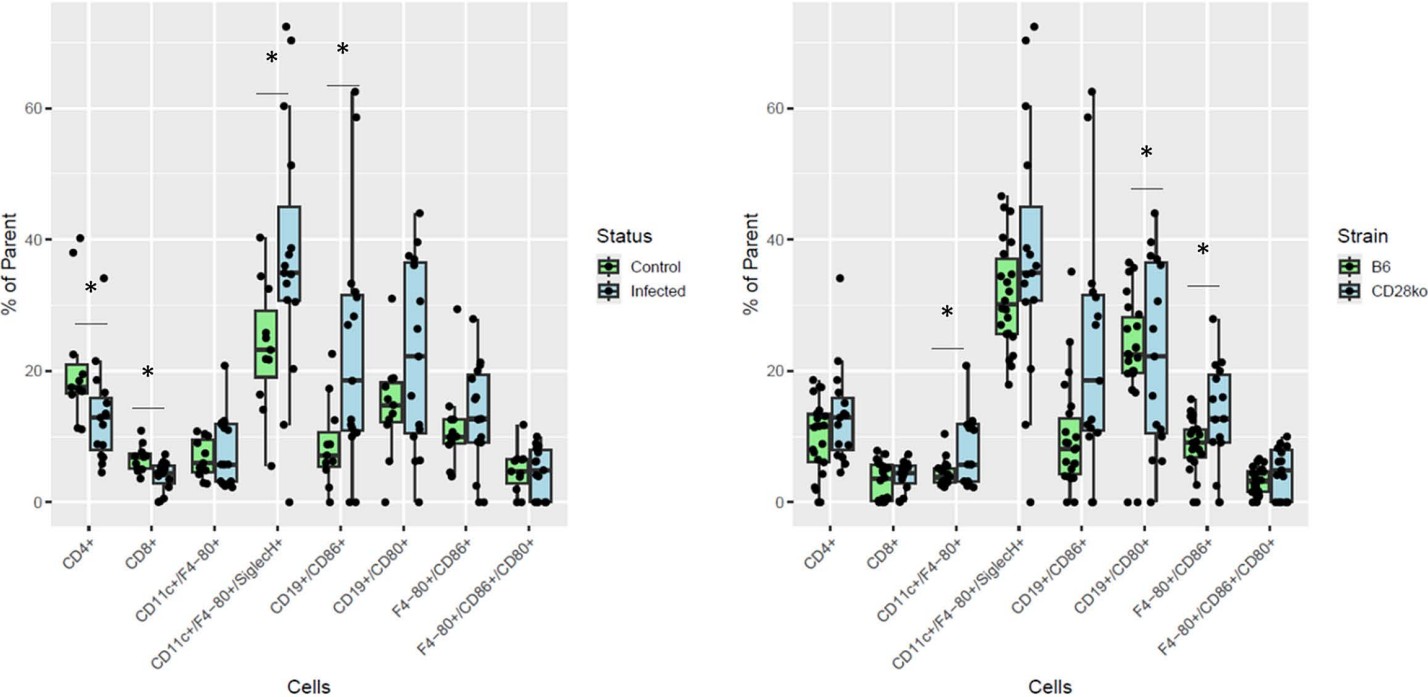

**A: control vs infected CD28ko mice**   **B: Infected B6 vs CD28ko mice**

**Fig 5. Multicolor flow cytometry analysis of immune cell populations in the spleens of B6 and CD28ko mice at four weeks post-infection. (A)** Infected CD28ko mice (N = 15) showed significantly lower levels of CD4+ but not CD8+T cells compared to uninfected controls (N = 11, p < 0.05). The plasmacytoid dendritic cell (pDC; SiglecH+) subset within the CD11c+F4-80+population was elevated in infected CD28ko mice compared to uninfected controls (p < 0.05). CD86+B cells (CD19+) were increased in infected CD28ko spleens compared to uninfected controls (p < 0.05). **(B)** Frequencies of CD4+ and CD8+T cells were comparable in infected B6 mice (N = 22) and infected CD28ko mice (N = 15, p > 0.05). CD11c+F4-80+cells were increased in infected CD28ko mice compared to B6 mice (p < 0.05). While no significant differences were observed in CD80+ or CD86+dendritic cells (CD11c+F4-80-) or macrophages (F4-80+CD11c-) between infected B6 and CD28ko mice, increased CD86-expressing macrophages (F4-80+CD11c-) and B cells (CD19+) were observed in infected CD28ko mice compared to B6 mice (p < 0.05). These findings highlight the contribution of CD4+ and CD8+T cells, as well as other antigen-presenting cells including macrophages and activated B cells, in controlling MmuPV1 infection in CD28ko mice.

elevated in the spleens of infected CD28ko mice compared to uninfected controls (**Fig 5A**, p < 0.05). While no significant differences were detected in dendritic cells (CD11c+F4-80-) expressing the CD28 ligands B7-1 (CD80+) or B7-2 (CD86+) between infected B6 and CD28ko mice, we observed an increase in CD86+ but not CD80+B cells (CD19+) and macrophages (F4-80+CD11c-) in CD28ko spleens. CD86-expressing B cells (CD19+) were also elevated in infected CD28ko mice compared to uninfected controls (**Fig 5A**, p < 0.05), indicating an immune response to viral infection in CD28ko mice. These findings further support the role of several key immune cells, including B cells (CD19+), and macrophages (F4-80+CD11c-), as well as their activated states in controlling MmuPV1 infection in the CD28ko mice.

### T-cell responses to MmuPV1 E6 and E7 in infected mice

T cell–mediated immune responses to the viral oncogenes E6 and E7 have previously been characterized in MmuPV1-infected mice after viral clearance [19,36]. To further investigate whether similar T cell-mediated responses were stimulated in our infected CD28ko mice, we used two reported MHC class I-restricted peptides: MmuPV1 E6/90–99 (KNIVFVTVR) and E7/69–77 (VLRFIIVTG) to stimulate spleen cells from orally infected CD28ko mice and analyzed

intracellular IFNγ expression [19]. A representative female (**Fig 6A**) and male (**Fig 6B**) CD28ko mouse shows the frequency of antigen-specific IFNγ-producing CD8+ T cells responding to the E6 epitope (KNIVFVTVR) and the E7 epitope (VLRFIIVTG). Slightly higher, but not statistically significant, levels of anti-E6 and E7 CD8+ T-cells were detected in orally infected CD28ko females compared to males (**Fig 6C**, p > 0.05). To determine whether these CD8/IFNγ dual positive cells

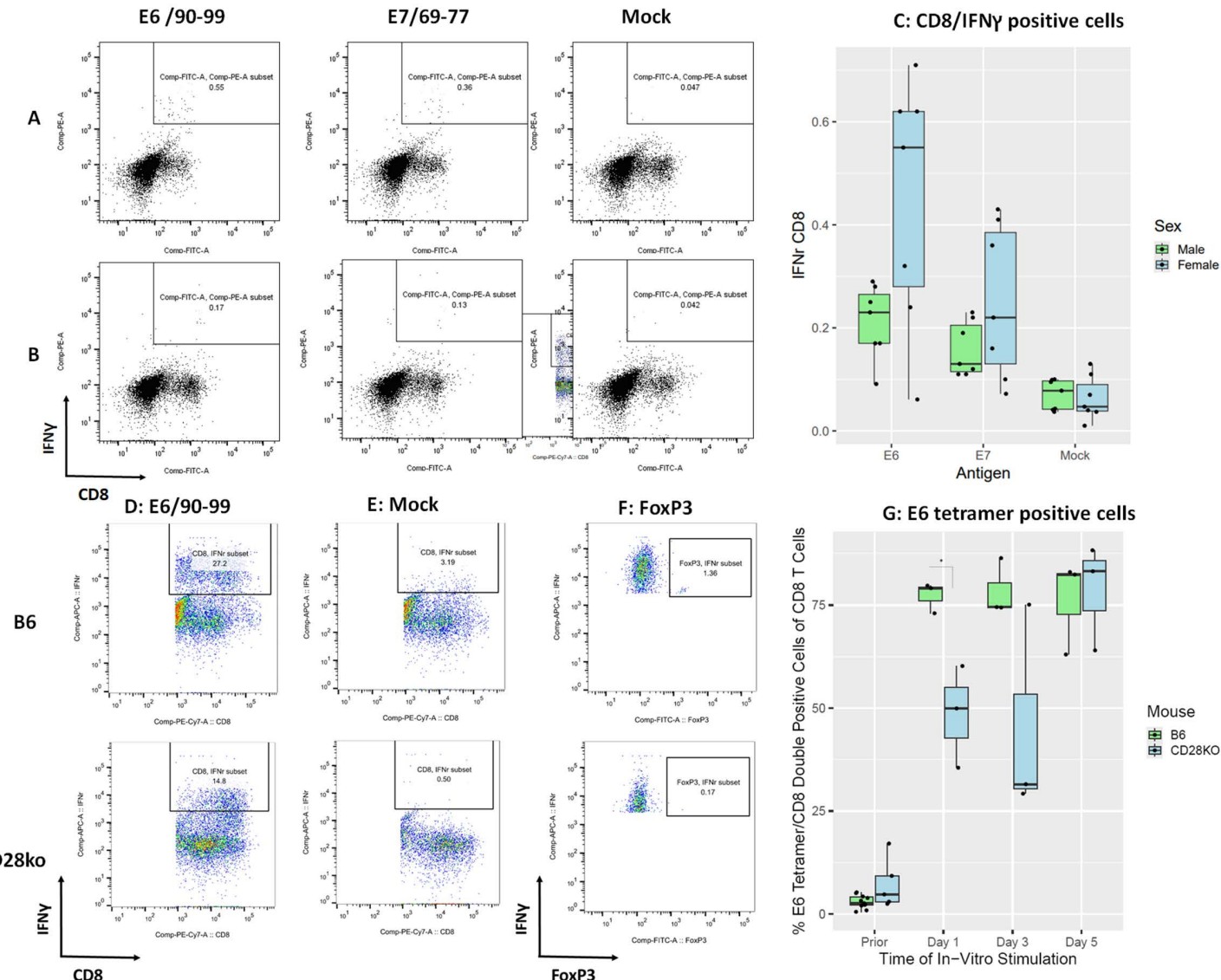

**Fig 6. A representative image of IFNγ-secreting CD8+T cells detected in CD28ko mice (N = 5/group, A female, B, male) after MmuPV1 infections.** Spleen cells (1 × 10^6) from infected CD28ko mice were stimulated with MmuPV1 E6 epitope (KNIVFVTVR), E7/69-77 (VLRFIIVTG), or mock, together with Brefeldin A for 4.5 hours before labeling with Alexa488 conjugated anti-mouse CD8α and PE conjugated anti-mouse IFNγ. Both anti-E6 and E7 CTLs were found in these mice with slightly more anti-E6 CTLs but not significant (C, p > 0.05). The CD8/IFNγ double positive cells were expanded in vitro (D) in both B6 and CD28ko mice. These cells were FoxP3 negative **(F)**. We also tested E6 specific tetramer positive CD8 T cells in infected B6 and CD28ko mice at different time points post in vitro stimulation. As shown in Fig 6G, extremely low levels of these tetramer positive cells were found in non-stimulated splenocytes from both B6 and CD28ko mice. On day one post stimulation, a substantial proportion of CD8 positive cells were also E6 tetramer positive. CD28ko mice showed a delay in expansion of these cells before five days post stimulation (G, *p < 0.05).

were regulatory T cells (Tregs), we performed FoxP3 co-staining and found nearly all CD8/IFNγ dual positive cells were FoxP3-negative, indicating that these CD8+ T cells do not have a regulatory phenotype (Fig 6D-6F).

To further evaluate the functional capacity of these T cells, we used a MHC tetramer synthesized by the NIH Tetramer Facility to determine whether CD28ko mice exhibit deficiencies in T cell expansion. Splenocytes from MmuPV1-infected B6 and CD28ko mice were stimulated with the E6/90–99 peptide and cultured for 1, 3, and 5 days. E6-specific tetramer-positive CD8+T cells were then quantified by flow cytometry [51,52]. As shown in Fig 6G (N = 5/group), only extremely low frequencies of tetramer-positive cells were detected in non-stimulated splenocytes from both MmuPV1-infected B6 and CD28ko mice. By day one post-stimulation, a substantial proportion of CD8+cells were E6 tetramer-positive in B6 mice. CD28ko mice exhibited a delayed initial expansion of E6-specific CD8+ T cells (Fig 6G, *p<0.05) but reached comparable frequencies of tetramer-positive CD8+ T cells by five days post-stimulation.

### Reduced in vitro cytotoxic activity of T cells from CD28ko mice

To further assess the potential of cytotoxic T lymphocytes (CTLs) to eliminate infected cells, we performed an in vitro killing assay using dendritic cells pulsed with either E6 peptide or mock peptide as target cells and labeled with 5 µM or 0.5 µM CFDA-SE, respectively. CD28ko CTLs exhibited significantly lower levels of target cell killing after in vitro expansion compared to B6 CTLs (Fig 7A, p<0.05), particularly at effector-to-target (E:T) ratios of 50:1 and 20:1.

Next, we conducted transcriptional analyses of two inflammatory cytokines IL-6 and TNFα, reported in previous human [53] and animal studies [31], to compare B6 and CD28ko CTLs. As expected, CD28 expression was significantly lower in CD28ko mice compared to B6 mice (Fig 7B, **p<0.01). Additionally, we detected significantly lower levels of TNFα but not IL-6 expression in CD28ko mice (Fig 7B, *p<0.05). CTLs typically produce TNF-α alongside perforin and granzyme B

**A: In vitro killing**

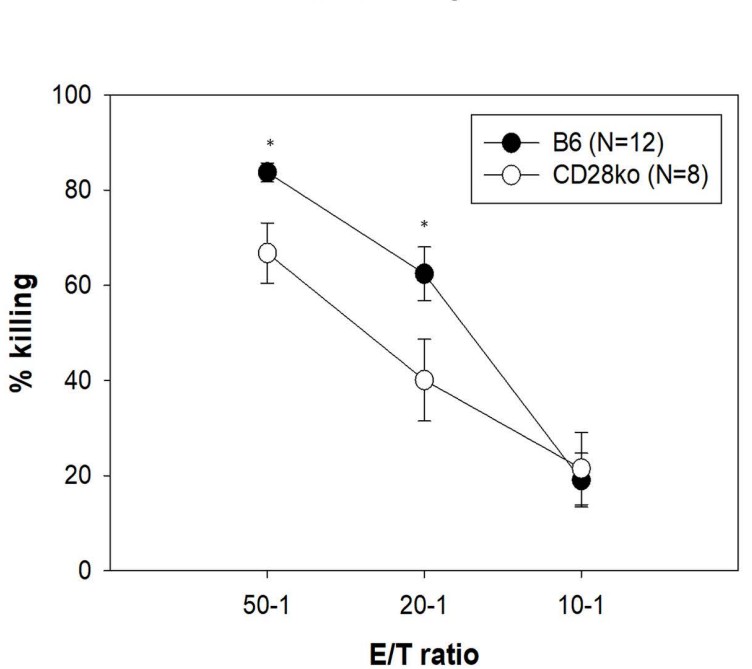

**B: Cytokine expression**

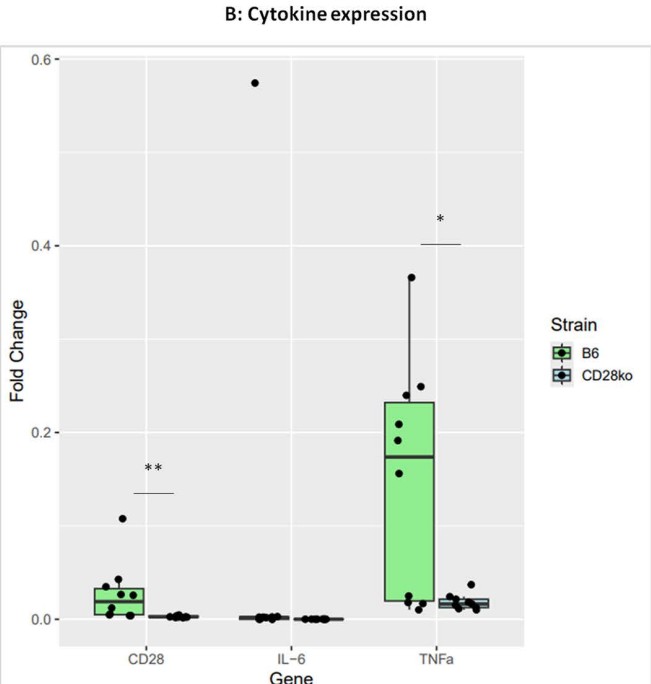

**Fig 7. Reduced cytotoxic activity and cytokine expression in CD28ko CTLs.** In vitro killing assay using dendritic cells pulsed with E6 or mock peptide and labeled with high (5 µM) or low (0.5 µM) CFDA-SE (N = 5/group). CD28ko CTLs showed significantly lower killing compared to B6 CTLs after expansion (p<0.05), particularly at E:T ratios of 50:1 and 20:1 (A, p<0.05). Transcriptional analysis showed markedly lower CD28 and TNFα but not IL-6 expression in CD28ko CTLs versus B6 CTLs, indicating impaired functionality that may contribute to delayed viral clearance (C, p<0.05).

during target cell killing, and TNF-α works synergistically with IFNγ to enhance antiviral and antitumor immunity by upregulating MHC class I and promoting immune cell recruitment [54,55]. Our findings therefore suggest a potentially compromised antiviral function, which may contribute to delayed viral clearance in these mice.

## Adoptive transfer of B6 CTLs but not CD28ko CTLs stimulated tumor regression in Rag1ko mice

We next evaluated whether adoptive transfer of equal numbers of these E6/90–99 tetramer positive CTLs from B6 and CD28ko mice could promote viral clearance and tumor regression in infected Rag1ko mice. Three groups of Rag1ko mice (N=4–7) were infected with $1 \times 10^8$ viral genome equivalents of MmuPV1 at both the tail and oral cavity, as previously described. Five weeks post-infection, E6/90–99 epitope-specific CTLs from either B6 or CD28ko mice, harvested five days after in vitro expansion, were administered via intravenous (IV) or intraperitoneal (IP) injection monthly in three doses. The control group (N=7) received no treatment. Tumor size and oral viral load were monitored throughout the study and measurements started around week five after the final treatment. Among the seven mice treated with B6 CTLs, tail lesions regressed over an eight-week period following the initial treatment (**Fig 8A**). In contrast, lesions in CD28ko CTL-treated mice (N=4) showed delayed regression and remained similar in size to those in the untreated control group during the same time frame when regression was observed in the B6 CTL-treated mice (**Fig 8B**). Tumor sizes were significantly smaller in B6 CTL-treated mice compared to both CD28ko-treated and control mice 5 weeks after the last treatment (**Fig 8C**, *p<0.05). Strikingly, no viral DNA was detected in oral swabs from either B6 or CD28ko CTL-treated mice, in contrast to the control group at all corresponding time points for recording tail lesions (W7 post treatment, **Fig 8D**, **p<0.001).

## Significantly higher numbers of dendritic cells and granulocytes but not macrophages were recruited to infected male tongues in CD28ko mice

HPV-associated oral infection and cancer exhibit a significant sex bias in the human population and have been implicated in our mouse model [19,56]. Consistent with our previous study, higher levels of viral transcripts were detected in

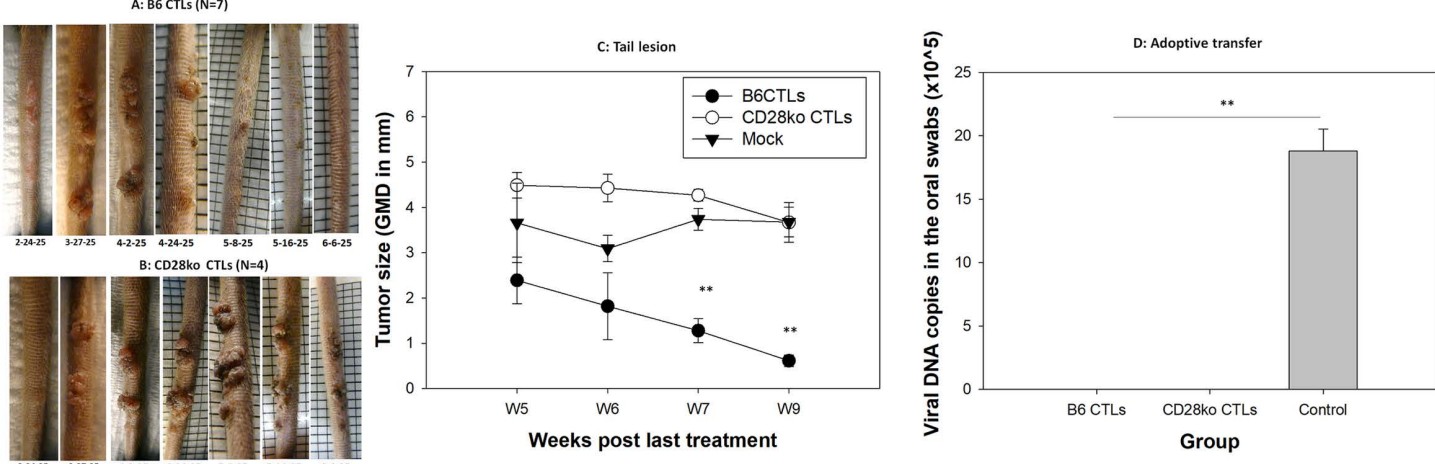

**Fig 8. Adoptive transfer of E6-specific CTLs promotes viral clearance and tumor regression.** Rag1ko mice (N=4–7 per group) were infected with MmuPV1 at tail and oral sites and treated five weeks later with three monthly doses of E6/90–99 tetramer+ CTLs from B6 or CD28ko mice via IV or IP injection; controls received no treatment. Tail lesions regressed within eight weeks in B6 CTL-treated mice (A), whereas CD28ko CTL-treated mice showed delayed regression similar to controls (B). Tumor sizes were significantly smaller in B6 CTL-treated mice compared to CD28ko and controls around five weeks after the last treatment (C, *p<0.05). Oral viral DNA was undetectable in both CTL-treated groups at all time points, unlike controls (D, **p<0.001).

the infected tongues of CD28ko males compared to females (**S4 Fig**), suggesting a sex-based difference in viral replication in the oral cavity of CD28ko mice [19]. We recently reported sex-related differences in several innate immune cell populations, including dendritic cells, macrophages, and granulocytes, in orally infected mice, which may be associated with viral control [19]. To determine whether similar sex differences exist in other immune cell profiles in the tongue and spleen at four weeks post-infection as shown in our previous study [19], we examined tissues that were harvested from both male and female CD28ko mice and analyzed using multicolor flow cytometry. Dendritic cells (CD11c$^+$), macrophages (F4-80$^+$), and granulocytes (Ly6G$^+$) in the spleen, as well as those recruited to infected tongues, were assessed using the same gating strategy described previously (**S2 Fig**). In brief, viable CD45$^+$ cells were gated based on CD markers to define immune cell populations in the spleens and tongues of infected mice. Two dendritic cell populations including lymphoid dendritic cells (CD11b$^{low}$CD11c$^+$, cDC1) and myeloid dendritic cells (CD11b$^{high}$CD11c$^+$, cDC2) were initially analyzed. Dermal DCs (CD205$^+$) of cDC2 and plasmacytoid DCs (pDCs, SiglecH$^+$) in cDC1 were further determined after excluding F4-80$^+$ macrophages. Based on the gating strategy of S2 Fig, a panel of myeloid immune cells including macrophages (F4-80+), granulocytes (Ly6G+), several dendritic cells: dermal DCs (CD205+), plasmacytoid DCs (SiglecH+), conventional DC1 and DC2 in the spleen (S5A and **S5C Fig**) and tongue (S5B and **S5D Fig**) of uninfected normal B6 and CD28ko mice based on sex were analyzed using the multiple color flow cytometry. Similar levels of myeloid cells were found in spleen between male and female B6 and CD28ko mice (A, p>0.05). Similar levels of most myeloid cells except dDCs in tongues of CD28ko males were significantly lower than those in B6 males (**S5B Fig**, p<0.05) suggesting CD28 absence in these mice did not dramatically change the population of these myeloid immune cells.

We further investigated these cells in the infected CD28ko mice based on sex. Higher numbers of cDC2 (CD11b$^{high}$CD11c$^+$) were found in infected tongues of males (N=4–5/group) when compared with females (**Fig 9A**, p=0.057). The percentage of dermal DC (dDCs, CD11c$^+$CD205$^+$) and plasmacytoid (pDC, CD11b$^{low}$CD11c$^+$SiglecH$^+$) cells was greater,

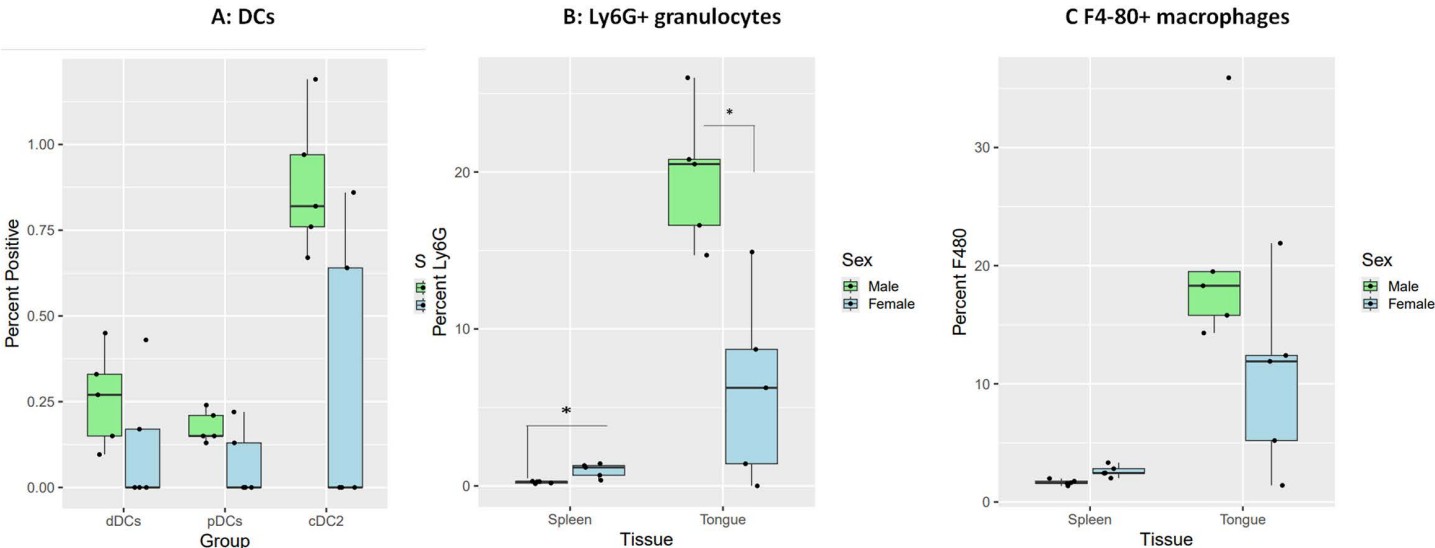

**Fig 9. Sex differences in immune cell populations in orally infected CD28ko mice.** Myeloid cells were distinguished using the gating strategy as shown in S2 Fig. Higher numbers of cDC2 (CD11b$^{high}$CD11c$^+$) were found in infected tongues of males (N=5) when compared with females (A, N=5/group, p=0.057). The percentage of dermal DC (dDCs, CD11c$^+$CD205$^+$) and plasmacytoid (pDC, CD11b$^{low}$CD11c$^+$SiglecH$^+$) cells was greater, but not statistically significant, in males when compared with females (A, p>0.05). A significantly greater percentage of Ly6G$^+$ granulocytes were found in the tongues of males compared to those in corresponding females (B, p<0.05). In contrast, significantly more Ly6G$^+$ granulocytes were found in the spleens of infected CD28ko female mice (B, p<0.05). Although greater percentages of macrophages (F4-80$^+$) were found in the tongues of infected CD28ko males compared to corresponding females, the difference was not statistically significant (C, p>0.05).

but not significantly, in males when compared with females (**Fig 9A**, p > 0.05). A significantly greater percentage of Ly6G+ granulocytes were found in the tongues of males compared to those in corresponding females (**Fig 9B**, p < 0.05). In contrast, significantly more Ly6G+ granulocytes were found in the spleens of infected CD28ko female mice (**Fig 9B**, p < 0.05). Although greater percentages of macrophages (F4-80+) were found in the tongues of infected CD28ko males compared to corresponding females, the difference was not significant (**Fig 9C**, p > 0.05).

## Discussion

Our study demonstrates that CD28 deficiency delays viral clearance and increases replication in both cutaneous and mucosal tissues, underscoring the critical role of the CD28-dependent pathway in adaptive immunity. Its absence reduces T-cell efficacy, contributing to persistent infection and increased tumor risk, as observed in humans [4,11]. In patients, cutaneous lesions are readily identifiable, whereas mucosal infections often go undetected, potentially contributing to overall disease burden. Although MmuPV1 lesions at cutaneous sites regressed after five weeks in CD28ko mice [10], infection persisted longer in the lower genital tract in our current study, consistent with findings from a previous report [16]. CD28/CD80/CD86 signaling is critical for effective T-cell responses, as antibody blockade of these ligands in B6 mice mimicked the CD28ko phenotype. Additional CD4/CD8 depletion further promoted tumor growth, underscoring the importance of T-cell immunity in CD28ko mice for protection from MmuPV1-induced lesions.

CD28ko mice exhibited delayed expansion and reduced cytotoxic function of E6-specific CTLs, along with lower CD4+ and CD8+ T-cell levels and delayed CD4+ infiltration into infected tissues. Infected tail tissues from CD28ko mice exhibited increased numbers of antigen-presenting cells expressing CD86 (B7-2) and dendritic cells (CD11c+) compared to B6 controls, indicating enhanced local antigen presentation capacity during infection. Increased recruitment of CD86+ and CD11c+ myeloid cells may suggest compensatory mechanisms for viral control or more inflammation due to delayed viral clearance in CD28ko mice. Separately, we observed a delay in CD4 T-cell recruitment to infected sites in CD28ko mice, which is consistent with the known requirement for CD28-mediated co-stimulation in promoting optimal T-cell activation and trafficking. These observations represent distinct aspects of the immune response rather than a direct mechanistic link. Additional studies are needed to determine whether these phenomena are functionally connected.

Furthermore, tissue-specific differences observed in adoptive transfer experiments using MmuPV1 E6-specific CTLs indicate that immune dynamics may vary across tissues and influence viral control [19,36], but that CTLs from CD28ko mice have reduced in vivo anti-tumor activity. These novel findings highlight the critical role of CD28-dependent immune dynamics in controlling MmuPV1 infection.

Previous studies suggested that CD4 or CD8 deficiency in B6 mice did not result in visible lesions in cutaneous tissues [37], but microscopic analysis revealed persistent infection after CD4 depletion [36]. In CD28ko mice, depletion of CD4+ or CD8+ T cells increased viral activity and lesion size, while dual depletion further prolonged persistence. Recovery of T-cell populations restored clearance, underscoring the importance of T-cell immunity. Similar patterns were observed in our CRPV model, where tumors regressed after immunosuppression ended [38]. Additionally, blockade of CD80/CD86 in B6 mice delayed clearance and reduced CD28 expression in local infected tissues, paralleling CD28ko findings and confirming the critical role of CD28-dependent pathways in papillomaviral control. Whether this is due to reduced recruitment of CD28-expressing cells to infected sites requires further investigation. Notably, CD28ko mice also showed increased susceptibility and viral reactivation in a mouse gammaherpesvirus-68 model [57].

In agreement with previous reports [16,19,36], MmuPV1 E6/90–99 is the immunodominant T-cell antigen in CD28ko mice. HPV16 E6 also has been identified as the dominant antigen in HPV patients [58]. In addition to reduced CD8/IFN γ T cell responses after in vitro stimulation, we also observed a delayed expansion of E6/90–99 tetramer positive CD8 T cells in CD28ko mice compared to B6 mice. These CTLs in CD28ko mice exhibited decreased TNF-α levels relative to B6 mice. Together, these findings suggest that CD28 deficiency impairs both cytotoxic and helper T-cell proliferation and expansion to MmuPV1 antigens [19], underscoring the critical role of CD28-dependent pathways in controlling viral

infection. Bulk transcriptional profiling of these CTLs could provide additional insights into their characteristics; therefore, we propose future studies using single-cell approaches (e.g., scRNA-seq with TCR-seq) to clarify pathway-level differences in cytotoxicity, trafficking, and co-stimulation. For example, other molecules such as CD40 and fms-like tyrosine kinase-3 ligand (Flt3L), expressed on antigen-presenting cells, also contribute to CD8+T-cell activation. We recently showed that Flt3L-deficient mice exhibit delayed viral clearance [59]. Additionally, other studies have demonstrated that CD40 signaling can compensate for CD4+T-helper cells in priming helper-dependent CD8+CTL responses [60].

Adoptive transfer experiments demonstrated that both CD28ko- and B6-derived CTLs effectively cleared oral infection. In contrast, cutaneous sites in both B6 and CD28ko recipients exhibited delayed clearance, with complete resolution observed in B6 CTL recipients beginning around week five after the final treatment. Complete clearance of cutaneous tumors by adoptive transfer of B6 CTLs was reported in a previous study; however, mucosal sites were not investigated [36]. Our previous infection studies showed that infection progressed more slowly in mucosal tissues of athymic mice, suggesting viral dynamics that may allow adoptively transferred CTLs to effectively eliminate infection in oral sites [23], whereas immunocompetent mice, including B6, exhibit rapid resolution of infection [19]. The observed tissue-specific differences likely reflect distinct immune dynamics. Oral mucosal tissues are highly vascularized [61] and enriched with professional antigen-presenting cells, such as dendritic cells and macrophages, which facilitate rapid antigen recognition and T-cell activation [19]. In contrast, cutaneous sites often present a more complex and less permissive environment, characterized by lower vascularity, a dense extracellular matrix, and distinct populations of resident immune cells, which may limit CTL trafficking and effector function [62]. Additionally, differences in the local cytokine milieu, chemokine gradients, and expression of adhesion molecules could further influence CTL recruitment and retention. These factors, along with potential differences in viral replication kinetics, will be explored in future studies.

The clinical heterogeneity observed in patients with CD28 deficiency, ranging from asymptomatic individuals to those with severe HPV-associated disease, can be reconciled with our findings that both CD28-dependent and CD28-independent mechanisms contribute to viral control. One possible explanation is the engagement of alternative costimulatory pathways, such as ICOS or 4–1BB, which can partially compensate for the absence of CD28 signaling and sustain T-cell activation and survival [12,63,64]. Additionally, tissue-specific immune dynamics may play a role; for example, mucosal sites enriched with professional antigen-presenting cells and higher baseline inflammation [61] may facilitate CD28-independent activation, whereas cutaneous sites [62] with lower vascularity and limited APC access could exacerbate the functional deficit. Genetic background and polymorphisms in cytokine or chemokine genes may further modulate the magnitude of compensatory responses, influencing disease severity [65,66]. Finally, differences in viral replication kinetics and immune evasion strategies across tissues could interact with these host factors to shape clinical outcomes [66,67]. These findings underscore the complexity of tissue-specific immunity and suggest that CD28-independent mechanisms are more effective in mucosal tissues but less so in skin, with implications for HPV persistence in immunosuppressed individuals [4,68,69]. Further studies will determine the roles of these molecules/ pathways in viral control in our model.

CD28 signaling is essential for multiple facets of CD4+T cell activation [70,71] including proliferation, survival [72,73], glucose metabolism [74] and migration [75,76]. In the absence of CD28 signaling, T cells fail to respond to their cognate antigen and become anergic [74,77]. Consequently, CD28-deficient mice exhibit reduced expansion of effector CD4+T cells and fail to form T helper type 1 (Th1) and T follicular helper (Tfh) cells after infection or immunization [47]. In addition to reduced CD4+T-cell levels in CD28ko mice, we observed delayed recruitment of CD4+T cells to infected tissues along with increased CD86+APCs and CD11c+ dendritic cells, suggesting robust local antigen presentation despite impaired co-stimulation. These findings reflect distinct aspects of the immune response rather than a direct causal relationship. Further studies are needed to determine whether the recruited cells are effector CD4+T cells and whether their function is impaired in CD28ko mice.

We identified an increase in CD11c+F4-80+ cells in infected mouse spleens. These cells represent a hybrid population with features of both dendritic cells (DCs) and macrophages, often referred to as monocyte-derived dendritic cells

(moDCs) or macrophage-like DCs. These cells typically emerge during inflammation or infection and play a critical role in antiviral immunity [78]. We observed an increased SiglecH$^+$ subset in this cell population in addition to increased CD86 expressing B cells (CD19$^+$) in infected CD28ko mice. Since these cells can present viral antigens, secrete type I interferons for antiviral defense, and recruit T cells, they may help CD28ko mice clear the infection [78,79]. In cancer, these cells can promote anti-tumor immunity by presenting antigens and producing pro-inflammatory cytokines, especially when polarized by therapies (e.g., docetaxel combined with TLR agonists) [79]. However, under certain conditions, they may also contribute to immune suppression and chronic inflammation, influencing tumor progression or regression depending on the microenvironment. The role of this specific cell population in viral infection and tumorigenesis especially in CD28ko mice warrants further investigation.

In conclusion, CD28 is a key T cell co-stimulator that drives effective immune responses to papillomavirus infection through CD80/CD86-mediated signaling. In this study, blocking CD80$^+$/CD86$^+$ cells in B6 mice mirrored the phenotype of CD28 knockout mice. Despite delayed clearance and increased susceptibility in CD28ko mice, all eventually resolved MmuPV1 infection, indicating compensatory CD28-independent pathways, as observed in humans [10]. These findings highlight tissue-specific immune control and underscore the need for local immune profiling to inform site-targeted therapies. Notably, sex differences in oral infection parallel patterns seen in HPV-associated disease and may relate to the male-to-female ratio in oropharyngeal cancer [80]. Our genetic and immunological data align with prior evidence that CD28 deficiency does not preclude HPV clearance, providing a foundation for future studies on HPV pathogenesis and immune regulation [19,81,82].

## Materials and methods

### Ethics statement

All animal studies were reviewed and approved by the Institutional Animal Care and Use Committee of The Pennsylvania State University College of Medicine (PROTO201800577, approved 2/18/2019; PRAMS200747110, approved 5/06/2019; PROTO202102089, approved 01/17/2022) and followed NIH guidelines for care and use of animals in this research.

### Animals and mouse papillomavirus (MmuPV1) infection

All work on mice was approved by the Institutional Animal Care and Use Committee of the Pennsylvania State University College of Medicine (PSUCOM), and all procedures were performed in strict accordance with guidelines and regulations. Infectious mouse papillomavirus (MmuPV1) was isolated from lesions on the tails of mice from our previous studies [23]. In brief, papillomas scraped from the tails of mice were homogenized in phosphate-buffered saline (1×PBS) with a Polytron homogenizer (Brinkman PT10–35) at maximum speed for three minutes while chilling in an ice bath. The homogenate was spun in a bench centrifuge at 10,000 rpm, the supernatant was decanted into Eppendorf tubes, and the suspension of the mouse papillomavirus was stored at -80°C in glycerol (1:1, V/V). C57BL/6, Rag1ko, CD28ko (6–8 weeks old) mice were originally obtained from Jackson Laboratories and bred and maintained in the animal core facility of PSUCOM. Both male and female CD28ko mice were housed (2–5 mice/cage) in autoclaved cages within sterile filter hoods and were fed sterilized food and water in the PSUCOM BL2 animal core facility. Mice (4–10 weeks old) were sedated i.p. with 0.1ml/10g body weight with ketamine/xylazine mixture (100 mg/10mg in 10 mls ddH$_2$O) before being pre-wounded at different sites including two cutaneous (tail and muzzle) and three mucosal (vaginal for female, anal, and oral for both sexes) as described previously [29]. Twenty-four hours after wounding, the mice were again anesthetized and challenged with infectious MmuPV1 (containing 1×10$^9$ viral DNA genomes/each infected site) at pre-wounded sites. Monitoring was conducted biweekly for infection at muzzle and tail and progress was documented photographically for each animal [30]. For all experiments, mice were inoculated with MmuPV1 at a standardized scarification site to ensure consistency across groups. The scarification site measured approximately 2 mm in width and 20 mm in length and was created on the tail using a sterile scalpel blade. The site dimensions were kept uniform

for all animals and experimental replicates. Virus inoculum was applied directly to the scarified area and allowed to absorb before animals were returned to their cages. Tumor sizes in Fig 8D were measured using a grid reference (1 mm squares) when photographing the three dimensions of the lesions. GMD refers to the geometric mean diameter (in mm), calculated as the cube root of (length × width × height), as described in our previous rabbit studies. This approach provides a more accurate representation of tumor size when tumors are not perfectly spherical. Lavage samples were collected from the mucosal sites as reported previously [23]. For most studies, CD28ko mice of both sexes were sacrificed at 4 weeks post viral infection. Additional groups of animals of both CD28ko and C57BL/6 were also sacrificed at week 2–5 post infection for time point comparison. The infected tissues were harvested for additional viral and in situ analyses.

## Mouse immune cell depletion via in vivo monoclonal antibody administration

The studies in which CD28ko mice were administrated anti-CD4 and anti-CD8 or anti-CD80/CD86 antibodies were conducted according to our previous study with some modifications [15]. Mice were injected with either 300 μg of anti-mouse CD4 (clone GK1.5) or 300 μg of anti-mouse CD8 (clone 2.43 against CD8a) or both I.P. in 200 μl 1 × PBS at day 1, 2, 3 and CD80/CD86 (100 μg anti–B7-1 (16-10A1; BioXCell) and 100 μg anti–B7-1 (GL-1; BioXCell) every 2 days for up to 1 week before pre-scarification, and once weekly until sacrifice of the mice. All mice were challenged with 10 μl (1 × 10⁹) MmuPV1 on day seven and sacrificed at week four post treatment. All infected mice were monitored for tumor growth at the two skin sites (muzzle and tail). All animals were sacrificed for viral detection during week three post infections except CD4/CD8. Dual depleted CD28ko mice were treated for up to seven weeks and followed up until week 12 post infection.

## Viral DNA and RNA copy number analyses

Linearized MmuPV1 genomic DNA was used for standard curve determination by a probe qPCR system. The primer pairs (5'GGTTGCGTCGGAGAACATATAA 3' and 5' CTAAAGCTAACCTGCCACATATC 3') that amplify E2 were used with the probe (5' 6-FAM-TGCCCTTTCA/ZEN/ GTGGGTTGAGGACAG 3'-IBFQ). Each reaction consisted of 9 μl of ultrapure water, 5pmol of each primer, 9μl of Brilliant III qPCR Master Mix (Agilent) and 2μl of DNA template. PCR conditions were initial denaturation at 95°C for 3 min, then 40 cycles at 95°C for 5 seconds and 60°C for 10 seconds and run on an Agilent AriaMx. Viral copy numbers were converted into equivalent DNA load using the formula 1ng viral DNA = 1.2 × 10⁸ copy numbers (http://cels.uri.edu/gsc/cndna.html). For lavage samples, 2 μl of the 50 μl DNA extracts of each lavage were used for qPCR [19]. For the tissue samples, 100ng total DNA was used for qPCR. Lavage samples from uninfected animals were included as negative controls (>40 cycles). All samples were tested in at least duplicates. Viral titers and positivity of samples were calculated according to the standard curve [29].

RNA and DNA were extracted from tissues by homogenizing in Trizol reagent (Life Technologies) using a Polytron homogenizer (Brinkman PT10–35) for 60s at speed 6. The protocol for Trizol reagent extraction of RNA followed by DNA was followed according to the manufacturer's instructions. Viral RNA (E1^E4 transcripts) was quantitated using primers 5'-TAGCTTTGTCTGCCCGCACT-3' and 5'-GTCAGTGGTGTCGGTGGGAA-3' and probe 5'FAM-CGGCCCGAAGACA ACACCGCCACG-3'TAMRA [30]. 200 ng of RNA was reverse transcribed using the RevertAid First Strand cDNA synthesis kit (Thermo-Fisher) and 2 μl cDNA was used in the qPCR analysis. 500nM of each primer and 250 nM probe was used with the Brilliant III qPCR kit (Agilent) with the following qPCR conditions- 95°C for 3 min, then 40 cycles at 95°C for 5 seconds and 60°C for 10 seconds on an Agilent AriaMx qPCR machine. No RNA control or samples with cycles of more than 40 were considered negative in calculation.

## In vitro viral infection and neutralization assay

A spontaneously immortalized mouse keratinocyte cell line (K38, a generous gift from Dr. Julia Reichelt [83], University of Newcastle, UK) was seeded at 1.5 × 10⁵ cells per well in E-media containing 5% FBS and nystatin and were cultured with mitomycin C-treated (Enzo Life Sciences) J2 3T3 feeder cells as previously described [30]. 1μl of viral extract from our MmuPV1

stock ($1 \times 10^8$ genome DNA equivalent) or supernatant from tissue homogenate was used for infection in K38 cells. For serum neutralization assay, the virus was incubated with different dilutions of supernatant at 37°C for 1 hour before adding to one well of a 12-well plate for each sample. The cells were incubated at 37°C for three additional days after the initial virus binding step and harvested with Trizol reagent (Life Technologies). Total RNA was extracted from the infected cells, and infectivity was assessed by measuring viral E1^E4 transcripts with qRT-PCR E1^E4-forward, 5′-CATTCGAGTC ACTGCTTCTGC-3′; E1^E4-reverse, 5′-GATGCAGGTTTGTCGTTCTCC-3′; E1^E4-probe, 5′-6-carboxyfluorescein (FAM)-TGGAAAACGAT AAAGCTCCTCCTC AGCC-6-carboxytetramethylrhodamine (TAMRA)-3′ as previously described with a few modifications as follows: The Brilliant III Master mix kit (Agilent) was used for the qRT-PCR reactions and carried out on an Agilent AriaMx. The following cycling conditions were applied: 50°C for 10 minutes (reverse transcription), 95°C for 3 minutes, and 40 cycles of 95°C for 5 seconds and 60°C for 10 seconds. The viral RNA copy numbers were calculated with a standard curve as described previously [30]. The neutralization rate was calculated based on several repeated experiments and compared to viral infection control alone.

## Multi-color flow cytometry

Immunophenotyping was performed by multi-color flow cytometry, as reported in a previous study [19], with mAbs from Biolegend against CD49b (pan-NK cells) (DX5), CD28 (E18, 37.51), CD3 (17A2), CD5 (53-7.3), CD4 (GK1.5), CD19 (6D5), CD20 (SA275A11), CD45 (30-F11), Ly6G (1A8), Ly6C (HK1.4), CD68 (FA-11), CD11c (N418), CD80 (16-10A1), CD86 (GL-1), CD8α (53-6.7), F4-80 (BM8), NK1.1 (PK136), CD11b (M1/70), IFNγ (XMG1.2), Siglec H (551), CD205 (NLDC-145), TCR-γδ. Cells were also stained with the Aqua Live/Dead Cell Stain Kit (Thermo Fisher Scientific). When required, cells were fixed and permeabilized with a fixation/permeabilization kit (eBioscience) or Perm buffer III (BD) after cell surface staining, for intracellular staining. Samples were analyzed using the LSR Fortessa in the Flow Cytometry Core of the Penn State College of Medicine (RRID:SCR_021134).

For multi-color flow cytometry, single viable leukocytes (CD45$^+$) from either spleens or tongues were gated for subset discrimination as described previously [19]. Different immune cell populations of viable CD45$^+$ cells were quantified from the spleens of both control and infected B6 and CD28ko mice. To compare immune cells expressing CD80 and/or CD86, CD45$^+$ cells were further separated into CD11c/F4-80 quadrants. pDCs (SiglecH$^+$) from both CD11c/F4-80 double-positive and CD11c$^+$/F4-80$^-$ populations were then analyzed for CD80 and CD86 expression. CD11c$^-$/F4-80$^+$ macrophages and CD19$^+$ B cells were also analyzed for CD80 and CD86 expression (Fig 5).

For additional assays to compare spleens or tongues of infected CD28ko mice, total dendritic cells (CD11c$^+$F4-80$^-$) by exclusion of macrophages (F4-80$^+$) were further analyzed to distinguish two dendritic cell populations including lymphoid dendritic cells (CD11b$^{low}$CD11C$^+$, cDC1) and myeloid dendritic cells (CD11b$^{high}$ CD11C$^+$, cDC2). Dermal DCs (CD205+) of total DCs and plasmacytoid DCs (CD11b$^{low}$ SiglecH$^+$) from cDC1 were further determined. Macrophages (F4-80$^+$) and granulocytes (Ly6G$^+$) were defined within the CD11b$^{high}$ cell population. Unstained cells were included to define the threshold separating positive populations from negative control cells. The numbers indicate the percentages of each cell population in the parent viable single CD45$^+$ cells (S3 Fig and Fig 9). All data were then analyzed with FlowJo 10.1r8 software.

## In vitro cytotoxic T cell culture and function assay

Mouse spleen cells were harvested during the termination of the experiments. Spleen cell suspensions were cultured in RPMI-1640 medium (Corning, 10–040-CV) supplemented with 10% FCS (hyclone, SH30070), 20mM HEPES (Gibo, 15630080), 1mM sodium pyruvate (Gibco, 11360070), 50μM 2-mercaptoethanol (Gibco, 21985023), 2mM l-glutamine (Gibco, 25030024) and 1% P/S (Gibco, 15140122) at $1 \times 106$ cells per ml. Splenocytes were stimulated with either 1ug/ml of MmuPV1 E6/90–99 (KNIVFVTVR) or E7/69–77 (VLRFIIVTG) peptides (synthesized by the core facility of Penn State College of Medicine) or no peptide for 4–6hrs in the presence of brefeldin A (GolgiPlug, BD Biosciences, 1/1000 final

concentration) at 37°C with 5% $CO_2$. Cells were stained with anti-mouse CD8α (Clone 53-6.7, Biolegend), permeabilized (Fixation/permeabilization buffer, eBioscience), and stained for intracellular IFNγ (clone XMG1.2, Biolegend), and FoxP3 (clone MF-14, Biolegend), and analyzed by flow cytometry on a BD LSR-Fortessa. We used the forward-scatter area or height (FSC-A or FSC-H) and the side-scatter area or height (SSC-A or SSC-H) for dead cell and debris removal. The forward-scatter height (FSC-H), width (FSC-W), side-scatter height (SSC-H), and width (SSC-W) were used to gate single cells. IFNγ-secreting cells were calculated by gating on the double positive CD8/ IFNγ population.

### E6 tetramer positive CD8 + T cells expansion in vitro

E6/90–99 (KNIVFVTVR)/H-2Kb tetramer was synthesized by the NIH tetramer core. Splenocytes from infected B6 and CD28ko mice were thawed and cultured in vitro and stimulated with cell-activation-cocktail (Biolegend,423301) or Dynabeads Mouse T-Activator CD3/CD28 for T-cell expansion and activation up to five days in complete RPMI 1640 medium (Corning, 10–040-CV) supplemented with 10% inactivated FBS (hyclone, SH30070), 20mM HEPES (Gibo, 15630080), 1mM sodium pyruvate (Gibco, 11360070), 50μM 2-mercaptoethanol (Gibco, 21985023), 2mM l-glutamine (Gibco, 25030024) and 1% P/S (Gibco, 15140122) with recombinant mouse IL-2 (BioLegend, 575406) at 100 IU/ml/$1 \times 10^6$ cells per ml with 1ug/ml of E6/90–99. E6 tetramer/CD8 dual positive cells were collected at days 1, 3, and 5 post in vitro stimulation. E6 tetramer positive cells from CD8+ T cells were analyzed by flow cytometry on a BD LSR-Fortessa using the same strategy shown above.

### In vitro killing and antiviral cytokine assay

Bone marrow-derived cells harvested from B6 mice were stimulated with complete RPMI 1640 medium (Corning, 10–040-CV) supplemented with 10% inactivated FBS (hyclone, SH30070), 20mM HEPES (Gibo, 15630080), 1mM sodium pyruvate (Gibco, 11360070), 50μM 2-mercaptoethanol (Gibco, 21985023), 2mM L-glutamine (Gibco, 25030024) and 1% P/S (Gibco, 15140122) and 10% mouse GM-CSF (supernatant of X6.3Ag8–653 cells) for five days for target cell production. $1 \times 10^4$ target cells cultured with 1ug/ml of E6/90–99 or a control peptide were incubated in either 5uM or 0.5uM carboxyfluorescein succinimidyl ester (CFSE, HY-D0938, MedChemExpress, NJ, USA) for 30 minutes, washed twice with medium and added to different dilutions of effector CTLs (E/T ratios of 50:1, 20:1 and 10:1) for up to six hours. The cells were harvested and detected by flow cytometry. The percentage of cells killed by CTLs was calculated using the control peptide-pulsed target cells as baseline.

Bulk CTLs cultured from both MmuPV1 infected B6 and CD28ko mice (N = 10–23) after in vitro E6/90–99 peptide stimulation for five days were tested for transcription of two antiviral cytokines, IL-6 and TNFα in addition to CD28, using previously described methods. In brief, CTLs were harvested in Trizol, and total RNA was extracted. 200ng of total RNA was reverse-transcribed into cDNA, followed by amplification using the Brilliant III SYBR Green Master Mix kit (Agilent) on an Agilent AriaMx. The following primers were used: IL-6 (sense 5'-AACGATGATGCACTTGCAGA-3', antisense 5'-GGTACTCCAGAAGACCAGAGG-3'); TNFα (sense 5'-AGCCCCCAGTCTGTATCCTT-3', antisense 5'-CTCCCTTTG-CAGAACTCAGG-3'); CD28 (sense 5'-GAAGGAACAGACTCCTTCAAAGTGA-3', antisense 5'-CTGGTAAGGCTTT CGAGTGAGC-3'); and β-actin (sense 5'-AGGGTCAGAAGGACTCCTACG-3', antisense 5'-GAGGTAGTCTGTCAG-GTCCC-3'). Cycling conditions were as follows: 95°C for 3 minutes, and 40 cycles of 95°C for 5 seconds and 60°C for 10 seconds. Relative fold changes in each transcript were calculated using the 2-ΔCT method (CT of each target gene normalized to β-actin) (Fig 7B).

### In vivo killing by adoptive transfer assay

Pooled CTLs from in vitro E6/90–99 peptide stimulation for five days were further tested for viral clearance and tumor resolution. Three groups of Rag1ko mice (N = 4–7) were infected with MmuPV1 (containing $1 \times 10^8$ viral DNA genomes/each infected site) at pre-wounded sites (tail and oral cavity). Five weeks post infection, mice were randomly divided into three

groups and treated with one of three doses ($6 \times 10^6$, $6 \times 10^6$, $1 \times 10^7$) of in vitro expanded CD8/E6/90–99 tetramer positive CTLs of either B6 (N = 7) or CD28ko (N = 4) mice either iv or ip monthly. One group (N = 5) without treatment served as the control (Fig 8). Oral swabs were collected biweekly for viral DNA quantitation. Lesions on the tail were monitored and recorded until week 9 post last treatment.

### In situ hybridization (ISH), RNA in situ hybridization (RNA-ISH), immunohistochemistry (IHC) for viral detection

Infected tissue biopsies were fixed in 10% neutral buffered formalin and embedded in paraffin. Adjacent sequential sections were cut for hematoxylin and eosin (H&E), *in situ* hybridization (ISH), RNA *in situ* hybridization (RNA-ISH), and immunohistochemistry (IHC) as described in previous studies [15,26,29,31]. For ISH, a biotin labeled 3913 bp EcoRV/BamH1sub genomic fragment of MmuPV1 was used as the probe for the detection of MmuPV1 DNA in tissues [23,24,27]. Access to target DNA was obtained following incubation with 0.2 mg/ml pepsin in 0.1N HCl at 37°C for 8 min. After thorough washing, the biotinylated probe was applied and heated to 95°C for 5 min to achieve dissociation of target and probe DNA. Re-annealing was allowed for 2 hours at 37°C. Target-bound biotin was detected using a streptavidin AP conjugate followed by colorimetric development in BCIP/NBT. Counterstaining for ISH was Nuclear Fast Red (American MasterTech, Inc.). Viral RNA expression was detected in formalin fixed; paraffin embedded (FFPE) tissues by RNAscope technology (Advanced Cell Diagnostics, Inc.) using custom probes that mapped within the second exon of the E1^E4 ORF (nt 3139–3419) RNAscope® 2.5 VS Probe- V-MusPV-E4 following the manufacturer's protocol. After hybridization the bound probes were detected by colorimetric staining using RNAscope 2.5 HD Assay – BROWN and counterstained with hematoxylin. For IHC, a rabbit anti-MmuPV1 E4 polyclonal antibody (Dr. John Doorbar) or anti-MmuPV1 L1 antibody (an in-house mouse monoclonal antibody MPV.B9) were used on FFPE sections. Detection was achieved using the ImmPRESS anti-rabbit IgG polymer system (Vector # MP-7801) or M.O.M. (Mouse on Mouse) ImmPRESS HRP (Peroxidase) Polymer Kit (Vector # MP-2400) using ImmPACT NovaRED Substrate (SK-4805). Before mounting, the slides were counterstained with 50% hematoxylin Gill's No. 1 solution (Sigma-Aldrich) and 0.02% ammonium hydroxide solution (Sigma-Aldrich) [32]. The mean intensity of viral E4 staining in each area was measured using ImageJ software and analyzed.

### Immunofluorescent (IF) analyses for local immune cells

Mouse frozen tissue (tail and spleen) sections were processed for immunofluorescent (IF) analyses. Sections were blocked by incubation for 1 h in PBS with 5% goat serum and incubated with primary antibodies at 4°C overnight. The following primary antibodies were diluted in PBS 5% FCS and used at the appropriate dilution: Alexa 488 conjugated rat monoclonal anti-mouse CD4 (1:50, Biolegend), rat monoclonal anti-mouse CD86 (1:50), rat monoclonal anti-mouse CD11c (1:50), rat monoclonal anti-mouse CD80 (1:50), rat monoclonal anti-mouse CD68 (1:50). Slides were mounted in Fluoromount-G (Clinisciences) supplemented with DAPI (1:1000, Cell Signaling). After 3 washes with PBS/T, secondary goat anti-mouse 488 (1:1000, Life Technologies) and DAPI (1:100) were added. Following 3 washes, coverslips were inverted and mounted onto glass slides with ProLong Gold Antifade Mountant (Invitrogen P36930). Images were taken by a wide-field fluorescence microscope (Zeiss Axio Imager Z2, Axiocam 506 mono) using Zeiss Zen Blue pro software. These images were further processed by Image J 1.8.0 (NIH, Bethesda, Maryland). The mean fluorescence intensities (MFI) in five fields of view of three sections of each mouse were calculated. To ensure accurate measurement of marker expression, we applied a rigorous thresholding and masking strategy to exclude nonspecific staining in the cornified layers. This approach was implemented to optimize the signal-to-noise ratio, as cornified layers often exhibit high background fluorescence that can artificially inflate intensity values. Thresholding and masking allowed us to differentiate true signal from background artifacts, ensuring that only biologically relevant staining was quantified. This method minimizes false-positive measurements and improves consistency across samples, thereby enhancing the reliability, reproducibility, and interpretability of the data. All images from samples of different groups were taken with identical microscope settings and adjusted uniformly.

## Statistical analyses

Viral load among different infected groups and between different sexes were analyzed by Wilcoxon rank sum [19]. Statistical significance between different immune cell populations, mouse strains, or different sexes was also tested using the same statistical program. Normalized viral E4 RNA copy numbers in Fig 1C were compared with the Kruskal-Wallis' test followed by Dunn post-hoc tests with a Bonferroni correction. Elsewhere, Wilcoxon rank sum tests were applied to compare quantitative variables in two groups, e.g., female vs. male. A Bonferroni correction was applied as needed, e.g., when performing two group comparisons at multiple anatomic sites. Statistical significance was assessed at the $\alpha = 0.05$ level. All analyses were performed with R 4.3.2 (https://www.R-project.org). Figures were created using the ggplot2 R package [84] or GraphPad Prism (10.2.3) or Sigma plot. The dunn.test R package (https://CRAN.R-project.org/package=dunn.test) was applied to perform Dunn post-hoc tests. In the relevant figures, n.s. indicates not significant; *$p > 0.05$.

## Supporting information

**S1 Fig. Viral E4 RNA (A and B, top panels) and E4 protein (A and B, bottom panels) detection in infected vaginal tissues of Rag1ko (A, panel), CD28ko (B, C) and C57BL/6 (B6) mice (not shown) with the basal layer marked in blue dotted lines.** Strong viral E1^E4 RNA (RNA-ISH, signals in brown, arrows) was detected in both Rag1ko (A) and CD28ko (B) infected vaginal tissues (two upper panels), but not in B6 mice (not shown) at four weeks post-infection. Viral E4 protein (signals in red, arrows) was also detected in the vaginal tissues of infected Rag1ko (A) and CD28ko mice (B) shown on the two bottom panels. Viral capsid protein L1 (signal in in red, arrows) was found in the infected vaginal tissues of CD28ko (C, top panel) but not in B6 (D, top panel). Ki67 (signal in red, arrows) was detected in CD28ko vaginal tissues (C, bottom panel), and a few signals were also found in B6 mice (D, bottom panel).
(TIF)

**S2 Fig. Immune cell profiling using multi-color flow cytometry assay.** The schematic flow chart of different immune cell populations from flow cytometry assays (A). Viable single leukocytes (CD45+) from either spleens or tongues were gated for subset discrimination. Total dendritic cells (CD11c+F4-80−) by exclusion of macrophage (F4-80+) cells were further analyzed to determine two dendritic cell populations including cDC1 (CD11b low CD11C+) and cDC2 (CD11b+ CD11c+). Higher but not significantly higher numbers of cDC2 were found in both B6 and CD28ko females (based on live cell population) when compared to corresponding males (B, $p > 0.05$, Wilcoxon rank sum). Dermal DCs (CD205+) of total DCs and plasmacytoid DCs (CD11b low SiglecH+) were further determined. Macrophages (F4-80+) and granulocytes (Ly6G+) were identified within the CD11b+ cell population.
(TIF)

**S3 Fig. Local antigen-presenting cells, including macrophages (F4-80+, A), B7-2+ cells (CD86+, B), and dendritic cells (CD11c+, C), were analyzed in infected tail tissues of B6 and CD28 ko mice at week three post-infection (N = 3/group).** Using mean fluorescence intensity (MFI) based on five field views of three whole sections, we observed similar levels of macrophages (F4-80+) in the infected tissues of B6 and CD28ko mice (S3A Fig, $p > 0.05$). However, increased expression of B7-2 (CD86+) and dendritic cells (CD11c+) on antigen-presenting cells was found in CD28ko mice compared to B6 mice (S3B Fig, $p < 0.05$; S3C Fig, $p < 0.05$). These findings suggest that enhanced recruitment of CD86+ cells and dendritic cells, or increased expression of these markers at infected sites, may compensate for delayed CD4+ T-cell infiltration and contribute to viral clearance in CD28ko mice.
(TIF)

**S4 Fig. Sex differences in viral transcripts in the tongues of MmuPV1 infected B6 and CD28ko mice.** B6 and CD28 knockout (KO) mice (N = 5–7 per group, including both males and females) were infected at the base of the tongue as previously described. At four weeks post-infection, tongues were harvested for quantification of viral E1^E4 transcripts.

Increased MmuPV1 E4 RNA transcripts were observed in infected tongues of CD28 KO mice compared to B6 mice (* $p < 0.05$). Additionally, slightly higher but not significantly higher RNA levels were detected in male tongues compared to females within the CD28 KO group (* $p > 0.05$).
(TIF)

**S5 Fig. Sex-based differences in a panel of myeloid immune cells in uninfected B6 and CD28ko mice.** Based on the gating strategy of S2 Fig, a panel of myeloid immune cells including macrophages (F4-80$^+$), granulocytes (Ly6G$^+$), several dendritic cells: dermal DCs (CD205$^+$), plasmacytoid DCs (SiglecH$^+$), conventional DC1 and DC2 in the spleen (A, C) and tongue (B, D) of uninfected normal B6 and CD28ko mice based on sex were analyzed using the multiple color flow cytometry. Similar levels of myeloid cells were found in spleens between male and female B6 and CD28ko mice (A, $p > 0.05$). Most myeloid cells were similar in tongues except dDCs of CD28ko males were significantly lower than those in B6 males (B, $p < 0.05$) suggesting CD28 absence in these mice did not dramatically change the population of these myeloid immune cells.
(TIF)

## Acknowledgments

We are grateful for the assistance of Dr. Jianhong Zhang and Joseph Bednarczyk at the Flow Cytometry Core of the Penn State College of Medicine RRID:SCR_021134. We thank the NIH Tetramer Core Facility (contract number 75N93020D00005) for providing MmuPV1 E6/90–99 (KNIVFVTVR)/H-2K$^b$ tetramer.

## Author contributions

**Conceptualization:** Jean-Laurent Casanova, Neil Christensen, Vivien Béziat, Jiafen Hu.

**Data curation:** Sarah Brendle, Jingwei Li, Song lu, Michael Kozak, Debra Schearer, Joshua Place, Karla Balogh, Jiafen Hu.

**Formal analysis:** Sarah Brendle, Jingwei Li, Song lu, Todd D. Schell, Vonn Walter, Jiafen Hu.

**Funding acquisition:** Adam D. Burgener, Thomas T. Murooka, Jiafen Hu.

**Investigation:** Sarah Brendle, Jingwei Li, Song lu, Michael Kozak, Vonn Walter, Joshua Place, Karla Balogh, Jean-Laurent Casanova, Neil Christensen, Thomas T. Murooka, Vivien Béziat, Jiafen Hu.

**Methodology:** Sarah Brendle, Jingwei Li, Song lu, Todd D. Schell, Michael Kozak, Debra Schearer, Joshua Place, Karla Balogh, Jiafen Hu.

**Project administration:** Jiafen Hu.

**Resources:** Todd D. Schell, Michael Kozak, Jean-Laurent Casanova, Adam D. Burgener, Thomas T. Murooka, Yusheng Zhu, Vivien Béziat, Jiafen Hu.

**Software:** Sarah Brendle, Todd D. Schell, Vonn Walter.

**Supervision:** Yusheng Zhu, Jiafen Hu.

**Validation:** Sarah Brendle, Jingwei Li, Song lu, Todd D. Schell, Michael Kozak, Vonn Walter, Debra Schearer, Karla Balogh, Jiafen Hu.

**Visualization:** Sarah Brendle, Jingwei Li, Song lu, Debra Schearer, Yusheng Zhu, Vivien Béziat, Jiafen Hu.

**Writing – original draft:** Jiafen Hu.

**Writing – review & editing:** Sarah Brendle, Song lu, Todd D. Schell, Michael Kozak, Vonn Walter, Joshua Place, Jean-Laurent Casanova, Neil Christensen, Adam D. Burgener, Thomas T. Murooka, Yusheng Zhu, Vivien Béziat, Jiafen Hu.

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
