## [Decision Letter · Decision Letter 0]

26 Jul 2024

Dear Dr. Hu,

Thank you very much for submitting your manuscript "CD28-deficient mice are vulnerable to mouse papillomavirus MmuPV1 infection of the skin and mucosae" for consideration at PLOS Pathogens. As with all papers reviewed by the journal, your manuscript was reviewed by members of the editorial board and by several independent reviewers. In light of the reviews (below this email), we would like to invite the resubmission of a significantly-revised version that takes into account the reviewers' comments.

Both reviewers 2 and 3 raised many major concerns that you need to address. These are not not simply organizational and language clarity concerns. There are concerns with not being provided clear indication of what you are interpreting your studies to mean.  How do these studies extend our understanding of the role of CD28 in papillomavirus pathogenesis on host immune responses to these viruses.  There were also concerns with statistical analysis being required, as well as better description of what your experiments were actually measuring (e.g. need for more accurate labeling of data, more accurate description in the results section). Importantly, you may need to  perform additional experiments to provide greater mechanistic insights not apparent from the experiments you described in the current version of your ms (e.g. more informative cytometric analyses to distinguish immune cell types, better assays for assessing immune cell functionality).  While there were recommendations for rejection of this ms,, I want to encourage you to use the constructive criticisms of the reviewers to improve the impact of your study.

We cannot make any decision about publication until we have seen the revised manuscript and your response to the reviewers' comments. Your revised manuscript will be sent back to reviewers for further evaluation.

Sincerely,

Paul F. Lambert

Academic Editor

PLOS Pathogens

Alison McBride

Section Editor

PLOS Pathogens

Michael Malim

Editor-in-Chief

PLOS Pathogens

orcid.org/0000-0002-7699-2064

Reviewer's Responses to Questions

**Part I - Summary**

Reviewer #1: This article submitted by Brendle et al continues to utilize the MmuPV1/mouse challenge model to explore the mechanisms underlying the immune control of cutaneous HPV, and the run-away infections seen in three patients with CD28 deficiency and in “treeman syndrome” in a subset of them. The study also continues to expand on their prior observation of elevated E4-specific IgG3 antibody responses in female mice after oral MmuPV1 challenge, and provides a more in-depth analysis in the immunophenotype of CD28 deficient versus wild type male and female mice, although most studies are correlative. The study suggests that additional factors are involved in the development of treeman syndrome beyond just CD28 loss.

Additional comments

Sentence in lines 62-65 needs re-writing.

Fig. 1C, incorrect statistical test

Supplementary Figs S2 and S3: Flow cytometry is a more quantitative approach to address potential differences splenocyte immunophenotype.

Line 170: Is there any evidence for sex-specific differences in antibody responses of patients, including isotype preferences, to HPV infection? Why do more men get Oropharyngeal cancers? Please provide some context/rationale here.

Does the antibody isotype difference simply reflect the total burden of genital disease being higher in the female mice (large surface area in the vaginal tract)?

Line 184, not really different strains (ko vs wt in same strain)

Has the greater IgG3 concentration in females been seen in mice challenged with other infectious agents? Please provide some more context about the biology of this isotypes versus others.

Reviewer #2: This manuscript reports that MmuPV1 infection in mice with CD28 KO develops warts in the skin, oral cavity, and anogenital tract. The authors found that MmuPV1-induced tumors regress within 5 to 6 weeks, and infected mice produce infectious virus in the vaginal lavage samples. This study follows up on their previous report of inherited T cell CD28 deficiency in human patients using mouse papillomavirus (MmuPV1) infection in CD28 KO mice.

While it provides some useful information, the manuscript is largely premature, revealing few new findings or mechanisms. It is expected that CD28 KO makes mice susceptible to MmuPV1 infection, as many previous studies have shown the importance of T-cell responses and that CD28 is required for T-cell activation.

Reviewer #3: In this study, the authors build on a recent collaborative report showing that humans with an inherited deficiency in T cell molecule CD28 are susceptible to skin HPV (types 2 and 4) infections and that C57BL/6 mice deficient for CD28 (CD28ko mice) were susceptible to MmuPV1 infections that produced tail and muzzle lesions. The CD28ko mice developed MmuPV1 warts at a similar rate as Rag-1-/- (mature T- and B-cell deficient) C57BL/6 mice, but warts regressed spontaneously in CD28ko mice whereas those in Rag-1-/- mice persisted (Béziat et al. 2021 Cell).

In the current study, the authors verified that CD28ko mice were susceptible to mmuPV1 lesion development at two cutaneous sites (muzzle and tail) and tested whether mucosal sites (oral, anal, and vaginal for females) were also susceptible to MmuPV1 infections. They report that cervicovaginal lavages of CD28ko mice produce infectious MmuPV1 virions. They find evidence that CD4+ and CD8+ T cells together help to control viral persistence in CD28ko mice. Higher levels of anti-E4 IgG3 antibodies were detected in CD28ko female mice (however, the importance of this or other antibody class differences is unclear). CD28ko mice were able to clear MmuPV1 infection at all tested epithelial sites, suggesting that CD28-independent pathways may play a role immune control. However, the study does not seem to provide much insight into the role of CD28 in controlling papillomavirus infections.

Overall, the study may provide some new insights into the control of papillomavirus infections. However, the manuscript is not written in a way that provides digestible information for a wide range of researchers from diverse infectious disease backgrounds. It is not always clear why certain viral and cellular targets were chosen for study over others. The claim that CD28 deficiency delays MmuPV1 clearance is not supported well by the data. The paper lacks convincing evidence that rigorous data analyses were performed. This makes it impossible for this reviewer to fully evaluate the results and their significance. The authors readily discuss their results and their implication in the context of their own prior work but are less prone to discuss how their findings mesh with results from other labs. It is also not clear how the findings of MmuPV1 infections at mucosal sites might relate to HPV infections in humans with CD28 deficiency, as these people have not been shown, to this reviewer’s knowledge, to have uncontrolled anogenital HPV infections.

**Part II – Major Issues: Key Experiments Required for Acceptance**

Reviewer #1: (No Response)

Reviewer #2: CD28 KO does not seem sufficient for MmuPV1 persistence, while Rag1 KO supports it. It will be interesting to investigate T cell responses and contributions of other immune cells in the absence of CD28. This will reveal an important mechanism of how warts regress after 5-6 weeks.

In Figure 4, it is strange why the authors depleted CD4 and CD8 T cells, although it is already known that these cells are required for MmuPV1 infection. Instead, it will be important to test if mice with depletion of the CD28 ligands CD80/86 support MmuPV1 infection in a similar manner to CD28 KO. Additionally, the efficiency of CD4 and CD8 T cell depletion should be confirmed since neutralizing antibody-mediated depletion cannot deplete all T cells, especially in the peripheral tissues. Wart formation in CD28 KO is also unclear in the images. The data should be quantified for appropriate comparison.

Immune cell profiling should be performed more thoroughly. CD11b and CD11c are particularly ineffective markers as they cannot identify specific types of myeloid cells. For example, CD11c is expressed in numerous immune cells, including DCs, monocytes, NK cells, neutrophils, and even some CD8 T cells. Consequently, the experiment provides no meaningful results beyond CD4 and CD8 T cells.

Intracellular staining of IFN-γ is not sufficient to determine antiviral T cell activity, as IFN-γ can be produced by immunosuppressive cells such as Treg and exhausted T cells. Thus, T cell functions should be analyzed using multiple assays, such as T cell proliferation and CTL assays.

Reviewer #3: The legend for Fig. 1C does not have an adequate description. What viral transcript is targeted? They y-axis is total viral mRNA in the entire lesion, or in a given amount of lesion or total RNA? There is the claim that “Viral RNA in infected cutaneous tissues were higher levels than those in the mucosal tissues” but only the tail and anus show significant differences. Another claim “Similar levels of viral RNA were detected at two cutaneous sites (tail and muzzle)” is not supported by the data -- they are not significantly different, but I do not think they can be characterized as similar.

In Fig. 2B, the y-axis should be labeled noting the amount of starting material and the limit of detection of the qPCR assay.

The in vitro neutralization data are not convincing (Fig. 3C).

References to Fig. 4 claim larger lesions and increased viral signals but no quantitative data from appropriately powered experiments are provided for either. The authors also claim “significantly smaller” tail lesions, but there are no measurements and no statistical analyses.

Fig. 5 legend does not specify the identity of TCRb, or demonstrate these tissues are indeed infected.

Similarly, in Fig. 6, the tissues require co-staining for MmuPV1 infection along with specifics of

quantification to support the claim that “significantly higher numbers of …. cells are identified in infected epithelium.” The y-axis is not adequately labeled (number of cells per area?). If this is mean fluorescence, it cannot provide numbers of cells.

Figs. 7 A-B, 8A-D, 9C, 10A-D should likely be shown as box and whisker plots and indicate the number of animals/samples in each group and show individual data point and number of replicates. Figs. 7C and 10A should indicate viral copies per volume or other unit of measure. It is not clear what “virus control” indicates. In Fig. 10A, were the tissues dissected to enrich for abnormal tissues?

Fig. 8: Why focus on E4 antibodies?

Fig. 9: What is the specific reason T cell mediated immune responses to E6 and E7 were characterized? What do the data reveal/imply? Same questions for Fig. 10.

Fig. 11. Does viral load refer to virions, viral genomes? This figure just summarizes the data, but does little to synthesize a model or explain any outcomes.

In Figs. S1, the basal layers should be marked for clarity. Please show and clearly label staining and infection sites for all strains of mice. It is not clear which panels include tissues from CD28ko mice.

Overall, there are incomplete data/information in the figures or figure legends for many experiments. A biostatistician should be added as an author to ensure that proper statistical analyses are performed for each experiment. The limits of detection for qPCR and RT-qPCR should we indicated on each graph. The number of individual data points and means (as appropriate) should be included for each graph. It is not clear how numbers of cells were counted (or whether fluorescence intensities were used). Were whole infected regions analyzed objectively, or were certain tissue regions selected subjectively?

Wording that should be modified:

Lines 57-58: Sentence implies HPVs are a single entity that can infect both cutaneous and mucosal epithelium and may all cause cancers. There are 100s of HPVs. Most HPVs appear to prefer either cutaneous or mucosal sites. Not all HPVs cause cancer.

Lines 59-60: To my knowledge, only genital HPV infections (and perhaps only at the cervix) have been studied in this detail. The specifics of this would be evident from primary literature, but the references for this statement are from reviews. The authors should minimize citing review articles.

For those less familiar with immune cell markers, the authors should clarify the roles of certain markers and their implications (e.g., in line 160: CD11c+, and in line 163: “CD86 … cells”)

Overall, there are numerous editing errors or inconsistencies. The English grammar needs attention in all sections. All citations should be checked as some references are obviously cited incorrectly (e.g., lines 413, 415), or cited non-numerically (e.g., line 469). Odd and inconsistent degree symbols (e.g., lines 471-472) and the notation for the K-38 cell line (lines 412, 416, 573) are used. PMID: 19907996. Misspelled or incorrect words, e.g., lines 63, 64, 513; Fig. 9: 1x109.

**Part III – Minor Issues: Editorial and Data Presentation Modifications**

Reviewer #1: (No Response)

Reviewer #2: Figures 1, 2, and 3 should be combined because Figures 2 and 3 showed the same conclusion, and Figure 1 is a control experiment for Figures 2 and 3.

For the same reason, Figures 5/6 and 7/8 also should be combined.

"we infected 18 mice (nine females and nine males) and 10 B6 mice (five females and five males)" - What are the first 18 mice if they are not B6?

The manuscript requires further editing to correct typos and errors.

Reviewer #3: (No Response)

PLOS authors have the option to publish the peer review history of their article (what does this mean? ). If published, this will include your full peer review and any attached files.

**Do you want your identity to be public for this peer review?** For information about this choice, including consent withdrawal, please see our Privacy Policy .

Reviewer #1: No

Reviewer #2: No

Reviewer #3: No
---

## [Decision Letter · Decision Letter 1]

10 Apr 2025

CD28-deficient mice are vulnerable to mouse papillomavirus MmuPV1 infection of the skin and mucosae

PLOS Pathogens

Dear Dr. Hu,

If you choose to submit a re-revised manuscript, then please submit it within 60 days Jun 09 2025 11:59PM. If you will need more time than this to complete your revisions, please reply to this message or contact the journal office at plospathogens@plos.org. Please include the following items when submitting your revised manuscript:

We look forward to receiving your revised manuscript.

Kind regards,

Paul F. Lambert

Academic Editor

PLOS Pathogens

Alison McBride

Section Editor

PLOS Pathogens

Editor-in-Chief

PLOS Pathogens

orcid.org/0000-0003-2946-9497

Editor-in-Chief

PLOS Pathogens

orcid.org/0000-0002-7699-2064

**Journal Requirements:**

1) We do not publish any copyright or trademark symbols that usually accompany proprietary names, eg ©,  ®, or TM  (e.g. next to drug or reagent names). Therefore please remove all instances of trademark/copyright symbols throughout the text, including:

- ® on page: 24 line 600

- TM on pages: 23 line 577, and 24 line 620.

2) We have noticed that you have uploaded Supporting Information files, but you have not included a list of legends. Please add a full list of legends for your Supporting Information files after the references list.

3) In the online submission form, you indicated that "The data supporting the finding of this study are available from the corresponding author upon request." All PLOS journals now require all data underlying the findings described in their manuscript to be freely available to other researchers, either

1. In a public repository

2. Within the manuscript itself

3. Uploaded as supplementary information.

4) Please amend your detailed Financial Disclosure statement. This is published with the article. It must therefore be completed in full sentences and contain the exact wording you wish to be published.

3) If any authors received a salary from any of your funders, please state which authors and which funders.

5) Please ensure that the funders and grant numbers match between the Financial Disclosure field and the Funding Information tab in your submission form. Note that the funders must be provided in the same order in both places as well. Currently, the order of the funders is different in both places. In addition, "Penn State Cancer Institute Program Project Development Award Sponsored by Highmark Community Health Reinvestment Fund" and "Canadian Institutes of Health Research" are missing from the Financial Disclosure field.

**Reviewers' Comments:**

Reviewer's Responses to Questions

**Part I - Summary**

Reviewer #2: The revised manuscript has been significantly improved, and most of the concerns have been clarified. However, a few issues still need to be addressed.

Reviewer #3: The authors verified that CD28ko mice were susceptible to MmuPV1 lesion development at two cutaneous sites (muzzle and tail) and tested whether mucosal sites (oral, anal, and vaginal for females) were also susceptible to MmuPV1 infections. While this is good to know, it is not surprising and does not provide any additional insight in MmuPV1 infections.

They find evidence that CD4+ and CD8+ T cells together help to control viral persistence in CD28ko mice. Higher levels of anti-E4 IgG3 antibodies were detected in CD28ko female mice (however, the importance of this or other antibody class differences remains unclear and is not well-linked to the CD28 focus of the study. Whereas CD28ko mice were able to clear detectable MmuPV1 infection at all tested epithelial sites, suggesting that CD28-independent pathways may play a role immune control, the study does not provide much insight into the role of CD28 in controlling PV infections.

The grammar and word/phrasing choices remain problematic.

PLOS Pathogens focuses on publishing groundbreaking research that significantly advances our understanding of pathogen biology. Although this study characterizes new MmuPV1 infection sites in CD28ko mice and provides some insight into the functions of CD28 in controlling MmuPV1 infections, the study is generally descriptive and only adds a bit to our knowledge about PV infections. I would not consider it to be groundbreaking or to significantly advance our understanding of pathogen biology.

Reviewer #4: Brendle et al, report on the phenotypic, virologic and immunological features of MmuPv1 infection in CD28KO/B6 background. They confirm the susceptibility of such CD28KO model to viral infection, present the features of skin, oral and anogenital infection in such model compared to B6 and Rag1 and investigate several immunological and virological features: presence of infectious virus in cervicovaginal lavage, effect of CD4/CD8 depletion or CD80/86 depletion on viral lesions, anti-E4 antibody titers and their sec differences, MuMPV-1 T cell responses, distribution of myeloid cells in infected tissue.

Despite the large amount of data presented, the manuscript does not provide a cohesive and unifying perspective on the immunological impact of CD28 KO on MmuPV1 and is not able to disentangle the role of CD28-dependent and independent antiviral responses, further clouding the understanding of the CD28KO B6 mice MmPV1 model.

**Part II – Major Issues: Key Experiments Required for Acceptance**

Reviewer #2: The data presented look solid. However, they are not sufficiently explained mechanistically. Are differential responses of T cells and macrophages in male and female CD28ko mice only in the B6 strain or also in human patients? It is also strange how E6-specific CD8 T cell numbers are increased (delayed though) in CD28ko mice in the absence of the critical costimulatory signaling. These issues should be thoroughly discussed.

Most flow cytometry images are difficult to read, and the labels are unclear. Also, it is necessary to quantify the data and indicate the number of repeats performed, along with their statistical significance.

Reviewer #3: As access to the basal layer is important for PV infection, it seems unlikely that the adequate depth or area of wounding of the different epithelial tissues (tail, muzzle, vagina, anus and tongue) can be controlled well enough to make strong conclusions about how well each tissue supports viral transcription. Since these comparisons are made, the text should include precise details about wounding, particularly the total area involved that receives 1e6 VGE MmuPV1. It simply does not seem possible to quantitatively compare infection at different epithelial sites without carefully controlled wounding.

Furthermore, although equivalent amounts of RNA were used for analyses (Fig. 1C), there is no way of knowing if precisely the same number of cells were infected in each sample. These issues make it impossible to determine how permissive each tissue is for PV infection.

It is unclear whether the MFI analyses, performed on selected fields of view microscopically are objective or subjective. This makes it difficult to determine whether the proper analyses have been done to determine the significance of the findings.

Results in Fig. 1K show both CD28ko and B6 mice lose MmuPV1 by week four. Thus, the infection does not persist longer in CD28ko mice - it does seem more robust during persistence.

Photos of the mouse tails from the top are not particularly revealing of the tumors (i.e., the lesions are not obvious). Accompanying side views might be helpful.

The low resolution in Fig. 4 (even in the high resol. figure) makes it impossible to interpret.

Fig. 7 shows ELISA data but the text indicates “significantly higher levels of viral transcripts were found in the infected tongues of CD28ko males when compared with those in females (Fig. 7A, p<0.05)” Lines 287-288. B-D should be cell staining?

Reviewer #4: 1. The current revised manuscript is following up on the original report of 3 human cases of a homozygous CD28 missense variant resulting in HPV2/4 cutaneous manifestations (without any mucosal or anogenital HPV lesions). Despite the lack of mucosal involvement in the human model and the significant heterogeneity in the clinical outcomes of the cutaneous infection, the authors focus in the current report on expanding MuMPV1 infection in different mucosal sites to clarify the immunopathogenesis in the context of CD28 deficiency. There are several elements which appear inconsistent and not well developed in the infection of different mucosal sites and would require further experimental validation:

A. Viral manifestations and immunological responses/landscape is most likely different in various mucosal sites and skin. The data presented in Figure 1 document presence of infection but does not really explore differences in viral replication/clearance in different sites. The kinetic of DNA replication in vaginal tissue in Figure 1K is very informative and does show that the slope of viral clearance between B6 and CD28KO is very similar after week 3 and that most of the differences in such tissue occur in the early timepoints of week 2-3. Is this observation consistent in other mucosal and cutaneous sites? Can the author provide such kinetics of viral DNA production in tongue and/or anus. Similarly, can a quantitative analysis of the kinetic of clearance of muzzle/tail lesions and/or viral DNA replication be provided and compared with the mucosal sites?

B. The kinetic analysis of viral replication should be used to inform the analysis of immune cells infiltration which in the current manuscript is limited only to the infected tail in Figure 4 (week 2 and week 4, single time point and not longitudinal kinetic). Also such analysis needs to utilize a quantitative imaging modality that can offer more accurate information on density of CD4 and CD8 T cells than the single field presented in figure 4. As the delay T cell infiltration is evoked as a key mechanism of viral persistence a longitudinal quantitative imaging of the tissue is essential to corroborate such claim.

2. The differences in level of Anti-E4 antibody between male and female CD28 KO is extensively investigated but is unclear if it correlates to any difference in the MmuPV1 lesions between male and female CD28 mice at cutaneous or anogenital sites (can test oral and anal infection in male and female to address mucosal along with 1 cutaneous site). In addition, if such differences are found to be relevant for the model (differences in lesion between male and female), the author should possibly investigate some phenotypic and functional features of B cells in B6 and CD28KO along with the differences in APC presented in figure 7 and S5. The authors can provide more robust data on MmuPv1 manifestations in male and female mice (may require larger n compared to the one provided so far to identify a significant difference or lack thereof) and then provide a unifying mechanistic insight into the immunological differences in male and female CD28KO and discuss why they are relevant for MmuPV1.

3. T cell expansion upon E6 peptide stimulation seems to be different in CD28KO vs B6 at early timepoints (Figure 6G). Such finding can be relevant and may recapitulate or be consistent with the delay in clearance of viral DNA (Figure 1K) and T cell infiltration (Figure 4) and could be further investigated/corroborated. The author can provide a bulk transcriptional analysis of their positive control T cells stimulated with anti CD3/CD28 as well as E6 tetramer positive T cells at day 1, 3 and 5 in B6 and CD28KO to investigate/corroborate functional defect of T cell function and CD28-dependent mechanism in their experimental model

4. CD28KO is associated with profound changes in the distribution of other immune cells (DC and granulocytes) at least in spleen and tongue. Did the authors check expression of CD28 in myeloid cells as well as in other immune cells which may be relevant for viral clearance in B6? Several authors have identified CD28 expression and function in other immune cells which may contribute to explain delay in viral clearance in the CD28KO/B6 model

- https://journals.aai.org/jimmunol/article/163/1/62/69482/Expression-of-a-Variant-of-CD28-on-a-Subpopulation and references therein: CD28 expression and function in murine and human NK

- https://pmc.ncbi.nlm.nih.gov/articles/PMC4744517/: CD28 expression on murine DC and its role as potent negative regulator of type 1 interferons

-https://onlinelibrary.wiley.com/doi/full/10.1002/eji.202048806. CD28 expression on macrophages.

**Part III – Minor Issues: Editorial and Data Presentation Modifications**

Reviewer #2: (No Response)

Reviewer #3: Again, Line 68 suggests that HPVs are a single entity that can cause cancers. There are 100s of HPVs. Not all HPVs cause cancer.

Line 189: what is a “regressive construct”?

Line 192: “The correlation of papillomavirus-associated lesions with increased immune cell infiltration including T cells and other immune cells has been demonstrated in human studies and other preclinical PV models [40-48]” Do the authors mean the correlation of lesion resolution?

Whereas the antibody findings are interesting, it remains unclear how they relate to the main goal of this study, which is to determine whether mucosal sites are susceptible to infection in CD28KO mice. I feel like the antibody data distract from the CD28 story or the antibody data need a more clear connection to CD28.

The English grammar still has many problems. One example of many includes “Comparable frequency of different DCs subsets, Ly6G +granulocytes, and macrophages (F4-80+ ) were found in CD45 + cells of spleens and tongues of B6 and CD28ko mice despite sex (Fig. S3A-D).” Despite sex? Encourage the authors to engage a scientific writer to edit the manuscript for word choices, clarity, and grammar.

Reviewer #4: (No Response)

PLOS authors have the option to publish the peer review history of their article (what does this mean? ). If published, this will include your full peer review and any attached files.

**Do you want your identity to be public for this peer review?** For information about this choice, including consent withdrawal, please see our Privacy Policy .

Reviewer #2: No

Reviewer #3: No

Reviewer #4: No

**Figure resubmission:**

**Reproducibility:**



---

## [Decision Letter · Decision Letter 2]

7 Aug 2025

CD28-deficient mice are vulnerable to mouse papillomavirus MmuPV1 infection of the skin and mucosae

PLOS Pathogens

Dear Dr. Hu,

Thank you for submitting your manuscript to PLOS Pathogens. After careful consideration, we feel that it has merit but does not fully meet PLOS Pathogens's publication criteria as it currently stands. Therefore, we invite you to submit a revised version of the manuscript that addresses the points raised during the review process.  It is important to take into consideration the concern raised by reviewer 4. The proposed experiment would have the potential to provide new mechanistic insight which is still perceived as lacking by some reviewers of your manuscript.

Please submit your revised manuscript within 60 days Oct 06 2025 11:59PM. If you will need more time than this to complete your revisions, please reply to this message or contact the journal office at plospathogens@plos.org. Please include the following items when submitting your revised manuscript:

We look forward to receiving your revised manuscript.

Kind regards,

Paul F. Lambert

Academic Editor

PLOS Pathogens

Alison McBride

Section Editor

PLOS Pathogens

Sumita Bhaduri-McIntosh

Editor-in-Chief

PLOS Pathogens

orcid.org/0000-0003-2946-9497

Michael Malim

PLOS Pathogens

orcid.org/0000-0002-7699-2064

**Journal Requirements:**

1) We do not publish any copyright or trademark symbols that usually accompany proprietary names, eg ©,  ®, or TM  (e.g. next to drug or reagent names). Therefore please remove all instances of trademark/copyright symbols throughout the text, including:

- ImmPACT® on page: 27.

2) In the online submission form, you indicated that The data supporting the findings of this study are available from the Penn State data repository upon request.. All PLOS journals now require all data underlying the findings described in their manuscript to be freely available to other researchers, either

1. In a public repository

2. Within the manuscript itself

3. Uploaded as supplementary information.

3) Please amend your detailed Financial Disclosure statement. This is published with the article. It must therefore be completed in full sentences and contain the exact wording you wish to be published.

4) Please ensure that the funders and grant numbers match between the Financial Disclosure field and the Funding Information tab in your submission form. Note that the funders must be provided in the same order in both places as well. Currently, the Financial Disclosure states there was no funding received.

**Reviewers' Comments:**

Reviewer's Responses to Questions

**Part I - Summary**

Reviewer #2: The manuscript has been improved, and I have no further major comments.

Reviewer #3: The authors previously demonstrated that patients with CD28 deficiency, a key T cell activator for adaptive immune responses (but a molecule that can also be expressed by mouse and human NK cells, mouse DCs, and macrophages), had HPV4-related severe recalcitrant warts and/or HPV2-related “tree man syndrome”. They also demonstrated a “causative effect” (a confusing term: a cause or effect, or both?) of CD28 deficiency for increased susceptibility to the mouse papillomavirus (MmuPV1) infection in tail skin tissues the MmuPV1 mouse model. Viral clearance in these lesions was observed, albeit with a delayed response.

The current study is based on the hypothesis that “mucosal sites will likewise be susceptible to MmuPV1 infection and will exhibit similarly delayed viral clearance.” Further, they state, “Therefore, after demonstrating cutaneous infection in CD28ko mice, we believe it is important to investigate whether CD28 deficiency influences viral infection across different tissue types.” However, the specific reason this might be important to yield valuable insight related to clinical observations or to provide increased understanding of papillomavirus pathogenesis is not made clear. This lack of guiding rationale is apparent in the manuscript that in many places seems to lack direction with a logical sequence of mechanistic questions that are addressed.

The authors have not been fully responsive to the comments in prior reviews.

Reviewer #4: The new set of experiments of in vitro killing assay and adoptive T cell transfer of CTL support some role of CD28 axis in the kinetic of clearance of viral lesion from cutaneous and mucosal sites. Although the study is not yet able to disentangle the role of CD28 -dependent and independent mechanisms for viral clearance and is still not clear whether the effect on viral lesions are mediated by altered function of the antigen-presentation, CD4 help, effector or regulatory functions, the work presented does provide sufficient novel insights into the role of cD28 signaling axis in viral control

**Part II – Major Issues: Key Experiments Required for Acceptance**

Reviewer #2: None

Reviewer #3: The current study is based on the hypothesis that “mucosal sites will likewise be susceptible to MmuPV1 infection and will exhibit similarly delayed viral clearance.” Further, they state, “Therefore, after demonstrating cutaneous infection in CD28ko mice, we believe it is important to investigate whether CD28 deficiency influences viral infection across different tissue types.” However, the specific reason this might be important to yield valuable insight related to clinical observations or to provide increased understanding of papillomavirus pathogenesis is not made clear. This lack of guiding rationale is apparent in the manuscript that in many places seems to lack direction with a logical sequence of mechanistic questions that are addressed.

The authors have not been fully responsive to the comments in prior reviews. For example, in most, if not all, figures with box/whisker plots, the numbers of tissues/replicates analyzed remains unstated. In Fig. S2, how many sections were quantified? Is it the entire tissue section? Does the MFI for the red also pick up the nonspecific staining appearing in the cornified layers? If not, how is this avoided? There seem to be positive CD86 signals in the epithelium of the tissues; were those also quantified?

What is the reason to refer to CD28 ligands CD80/CD86 without stating the cell types that could be expressing these proteins?

Without quantitative data from multiple sections and multiple mice, the information in Figure 4 is not compelling. Furthermore, the week 4, CD28 KO photo shows double positive cells primarily in what looks to be a hair follicle, and not clearly in the tumor.

The authors should explain why increased numbers of antigen presenting cells expressing B7-2 (CD86+ ) and dendritic cells (CD11c+) found in the infected tail tissues of CD28ko vs. B6 mice leads to the conclusion that suggest that delayed viral clearance in CD28ko mice correlates with a delay in recruitment of CD4 T cells to infected sites. My sense is that these two are not directly related, but the construction of the paragraph and the order of the sentences makes it seem the authors feel they are related. This is a good example of how the paper is confusing.

The rational for investigating antibodies remains absent in the context of CD28. This issue adds to the confusion of the paper. If there is a direct link between CD28 and E4 or other antibodies, it needs to be fully explained. If there is no or little link, these data should be removed from the paper to increase the clarity This issue has come up in every review iteration and it remains inadequately addressed.

The MmuPV1 inoculation is not adequately described, and this information was requested in the last review. What was the width x length of the scarification site? Were they the same in all the experiments and Fig. 1I, J?

It is not clear how the tumor sizes were measured in Fig. 6D; GMD is not defined.

As noted in the prior review, Fig. 7 shows ELISA data but the text indicates “significantly higher levels of viral transcripts were found in the infected tongues of CD28ko males when compared with those in females (Fig. 7A, p<0.05)” All of Fig. 7 is devoted to antibody data, not immune cell detection as the text indicates (should they all be Fig. S2?). Fig. 7 and Fig. S3 are the same.

Figs. S5-7 should display data as box-whisker or violin plots.

The English grammar remains problematic as noted in the last two reviews.

Line 68: The authors continue to refer to HPVs as a single entity, which is incorrect and misleading. How about: “HPVs are ubiquitous and high-risk HPVs types cause roughly 5 % of all human cancers.”

Reviewer #4: Bulk transcriptional analysis of the sorted tetramer+E6 CTL from B6 vs CD28KO may provide some additional mechanistic insights into qualitative difference of CTL form these 2 different conditions and their different kinetic/effect on in vitro killing assay and in vivo adoptive T cell transfers

**Part III – Minor Issues: Editorial and Data Presentation Modifications**

Reviewer #2: Most flow cytometry images are still very difficult to read, as the labels are small and blurry.

Reviewer #3: Figs. S5-7 should display data as box-whisker or violin plots.

Line 68: The authors continue to refer to HPVs as a single entity, which is incorrect and misleading. How about: “HPVs are ubiquitous and high-risk HPVs types cause roughly 5 % of all human cancers.”

Reviewer #4: 1. Specify at which timepoint the oral swabs were assayed for viral DNA in the adoptive T cell transfer experiments and discuss more extensively the difference between `delayed clearance in cutaneous sites` in CTL from CD28KO and complete and effective clearance of all oral virus DNA in both CD28KO and B^ WT

2. fig. 6 data :the clearance of tail lesions is assayed in the interval 5-9 weeks post-last treatment but it appears that in earlier time points there could be an effect of adoptively transferred CTL as already at week 5 post-last treatment there is a significant difference between mock and CD28ko compared to B6 (~2.5 mm GMD tumor size vs 4-5 mm). Data on the measures of the tail lesions at early timepoints post-last treatment or post-first and second monthly treatment can help understanding the kinetic of clearance and the differences among the experimental conditions. These data are important to infer an intrinsic defect of CTL function or their expansion and homeostasis in vivo.

3. the authors can discuss how the clinical heterogeneity of CD28 human deficiency can also be reconciled with the prominent role for both CD28 dependent and independent mechanisms in control of viral lesions in the mouse model

PLOS authors have the option to publish the peer review history of their article (what does this mean? ). If published, this will include your full peer review and any attached files.

**Do you want your identity to be public for this peer review?** For information about this choice, including consent withdrawal, please see our Privacy Policy .

Reviewer #2: **Yes:** Dohun Pyeon

Reviewer #3: No

Reviewer #4: No

**Figure resubmission:**

**Reproducibility:**



---

## [Editor Report · Decision Letter 3]

2 Feb 2026

Dear Dr. Hu,

We are pleased to inform you that your manuscript 'CD28-deficient mice are vulnerable to mouse papillomavirus MmuPV1 infection of the skin and mucosae' has been provisionally accepted for publication in PLOS Pathogens.

Best regards,

Richard B.S. Roden

Academic Editor

PLOS Pathogens

Alison McBride

Section Editor

PLOS Pathogens

Sumita Bhaduri-McIntosh

Editor-in-Chief

PLOS Pathogens

orcid.org/0000-0003-2946-9497

Michael Malim

Editor-in-Chief

PLOS Pathogens

orcid.org/0000-0002-7699-2064
---

## [Editor Report · Acceptance letter]

Dear Dr. Hu,

We are delighted to inform you that your manuscript, "CD28-deficient mice are vulnerable to mouse papillomavirus MmuPV1 infection of the skin and mucosae," has been formally accepted for publication in PLOS Pathogens.

Best regards,

Sumita Bhaduri-McIntosh

Editor-in-Chief

PLOS Pathogens

orcid.org/0000-0003-2946-9497

Michael Malim

Editor-in-Chief

PLOS Pathogens

orcid.org/0000-0002-7699-2064